# Intragenomic conflicts with plasmids and chromosomal mobile genetic elements drive the evolution of natural transformation within species

**Fanny Mazzamurro**[1,2]*, **Jason Baby Chirakadavil**[3], **Isabelle Durieux**[3], **Ludovic Poiré**[3], **Julie Plantade**[3], **Christophe Ginevra**[4], **Sophie Jarraud**[4], **Gottfried Wilharm**[5], **Xavier Charpentier**[3‡]*, **Eduardo P. C. Rocha**[1‡]*

**1** Institut Pasteur, Université Paris Cité, CNRS UMR3525, Microbial Evolutionary Genomics, Paris, France, **2** Collège Doctoral–Sorbonne Université, Paris, France, **3** CIRI, Centre International de Recherche en Infectiologie–Inserm, U1111, Université Claude Bernard Lyon 1, CNRS, UMR5308, École Normale Supérieure de Lyon, Univ Lyon, Villeurbanne, France, **4** Centre national de Référence des Légionelles–Centre de biologie Nord, Lyon, Cedex 04, France, **5** Robert Koch Institute, Project group P2, Wernigerode Branch, Wernigerode, Germany

‡ These authors are co-senior authors on this work.
* fanny.mazzamurro@pasteur.fr (FM); eduardo.rocha@pasteur.fr (EPCR)

## Abstract

Natural transformation is the only mechanism of genetic exchange controlled by the recipient bacteria. We quantified its rates in 786 clinical strains of the human pathogens *Legionella pneumophila* (Lp) and 496 clinical and environmental strains of *Acinetobacter baumannii* (Ab). The analysis of transformation rates in the light of phylogeny revealed they evolve by a mixture of frequent small changes and a few large quick jumps across 6 orders of magnitude. In standard conditions close to half of the strains of Lp and a more than a third in Ab are below the detection limit and thus presumably non-transformable. Ab environmental strains tend to have higher transformation rates than the clinical ones. Transitions to non-transformability were frequent and usually recent, suggesting that they are deleterious and subsequently purged by natural selection. Accordingly, we find that transformation decreases genetic linkage in both species, which might accelerate adaptation. Intragenomic conflicts with chromosomal mobile genetic elements (MGEs) and plasmids could explain these transitions and a GWAS confirmed systematic negative associations between transformation and MGEs: plasmids and other conjugative elements in Lp, prophages in Ab, and transposable elements in both. In accordance with the hypothesis of modulation of transformation rates by genetic conflicts, transformable strains have fewer MGEs in both species and some MGEs inactivate genes implicated in the transformation with heterologous DNA (in Ab). Innate defense systems against MGEs are associated with lower transformation rates, especially restriction-modification systems. In contrast, CRISPR-Cas systems are associated with higher transformation rates suggesting that adaptive defense systems may facilitate cell protection from MGEs while preserving genetic exchanges by natural transformation. Ab and Lp have different lifestyles, gene repertoires, and population structure. Nevertheless, they exhibit similar trends in terms of variation of transformation rates and its

**Data Availability Statement:** All relevant data are within the paper and its Supporting Information files.

**Funding:** This work was supported by the French Agence Nationale de la Recherche (ANR), under grant ANR-20-CE12-0004 (TransfoConflict). XC lab if funded by a grant Equipe FRM (Fondation pour la Recherche Médicale) EQU202303016268. EPCR lab is funded by a grant Equipe FRM (Fondation pour la Recherche Médicale) EQU201903007835 and by the Laboratoire d'Excellence IBEID Integrative Biology of Emerging Infectious Diseases [ANR-10-LABX-62-IBEID]. The funders had no role in study design, data collection and analysis, decision to publish, or preparation of the manuscript.

**Competing interests:** The authors have declared that no competing interests exist.

**Abbreviations:** ASEM, agarose soluble extract media; CFU, colony-forming unit; GWAS, genome-wide association study; HGT, horizontal gene transfer; ICE, integrative conjugative element; LMM, linear mixed model; LU, luminescence unit; MGE, mobile genetic element; RLU, relative luminescence unit; SNP, single-nucleotide polymorphism; ssDNA, single-stranded DNA; wGRR, weighted gene repertoire relatedness.

determinants, suggesting that genetic conflicts could drive the evolution of natural transformation in many bacteria.

## Introduction

Natural transformation consists in the uptake of exogenous DNA from the external surroundings of bacteria and its integration in their chromosome by homologous recombination. Unlike conjugation and transduction, 2 other key mechanisms of horizontal gene transfer (HGT), natural transformation is not encoded by mobile genetic elements (MGEs). It is the only known mechanism of HGT encoded and under the direct control of the recipient bacteria. It is the mechanism behind the "transforming principle" that resulted in the discovery of HGT [1] and ultimately to the identification of DNA as the material of genetics [2]. Transformation requires a transient physiological state, the "competence" state, during which the machinery necessary for DNA import and integration in the chromosome is expressed [3]. Relatively few species have been demonstrated to be naturally transformable, but many more encode the necessary machinery and it is suspected that they are also transformable [4]. The process starts by the capture of exogenous DNA by an extracellular type IV pilus (Pil genes), whose retraction conveys the DNA at the cell surface [5]. In *Helicobacter* this step depends on a machinery derived from type 4 secretion system [6]. A nuclease converts the DNA into single-stranded DNA (ssDNA) before its transport to the cytoplasm by ComEC [3]. Once in the cytoplasm, the incoming ssDNA is protected from degradation by DprA, which also recruits RecA [3]. The recipient homologous recombination machinery is then involved in genetic exchanges with the bacterial chromosome. In diderms, the ComM protein has a key role in this process, facilitating genetic exchanges between long heterologous DNA sequences [7].

Even if competence for natural transformation was discovered almost a century ago, the reasons for its existence are still debated [8] (S1 Fig). They include the promotion of allelic recombination [8], the acquisition of nutrients [9], and the uptake of DNA for repair [4]. The impact of transformation is significant. Thanks to homologous recombination, transformation can break co-adapted gene complexes and thus increase the efficacy of natural selection. Transformation also allows bacteria to acquire new functions that can be of adaptive value. For example, transformation enabled the acquisition of antibiotic resistance determinants by *Campylobacter jejuni* [10], *Streptococcus pneumoniae* [11], and *Acinetobacter baumannii* [12]. Recently, it was proposed that transformation eliminates deleterious MGEs by recombination in the flanking chromosomal core genes [13]. This chromosome-curing hypothesis implicates the existence of intragenomic conflicts between MGEs and the host regarding natural transformation. Accordingly, MGEs of *V. cholerae* (an integrative conjugative element, ICE) and of *C. jejuni* (a prophage) encode DNases that prevent transformation [14,15]. Many other MGEs insert and disrupt key competence genes [13]. Finally, several components of the transformation pathway are involved in other functions (e.g., adhesion and virulence with the Type IV pilus and homologous recombination with RecA) and pleiotropic interactions could contribute to the maintenance of transformation in natural populations. It is possible that several of these hypotheses contribute to selection for natural transformation.

The core components of the DNA uptake system and of homologous recombination are widely conserved across bacteria. Yet, large variations of transformation frequencies have been observed and many species are described as non-transformable even though they encode all necessary components. For example, *Pseudomonas stutzeri* is transformable, whereas

*Pseudomonas aeruginosa* is widely viewed as non-transformable [16]. Differences in transformability within species have also been observed. In well-established transformable species such as *S. pneumoniae*, *P. stutzeri*, and *Haemophilus influenzae*, from 30% to 60% of the isolates [17–19] consistently failed to transform. Extensive variations were also reported in wildlife, clinical and human isolates of *Acinetobacter baumannii* [20] and in clinical isolates of *Legionella pneumophila* [21]. The reasons of such large within-species variations in transformation frequencies remain poorly understood. Of note, hypotheses explaining the existence of transformation do not necessarily explain large variations of their rates within species. For example, one does not expect huge variations in transformation between strains under similar growth conditions when there is selection for DNA repair or use of DNA as a nutrient. This does not imply that such hypotheses are incorrect, yet suggests that additional forces are at play. Notably, if intra-genomic conflicts affect transformation rates then low transformation rates could evolve even if they are deleterious to the bacterium [21].

The previously cited studies used relatively small samples (fewer than 150 strains and sometimes a few dozens), which precludes the use of powerful statistical methods to understand the genetic basis of phenotypic variation. Here, we characterize the transformation rates and identify their genetic determinants in 2 phylogenetically distant *Gammaproteobacteria* species, *L. pneumophila* (Lp) and *A. baumannii* (Ab). These are important pathogens with different characteristics. Ab is one of the most worrisome antibiotic-resistant nosocomial pathogens [22]. Some strains are now resistant to nearly all antibiotics [23], an evolutionary process driven by chromosomal recombination events (possibly resulting from transformation) and by MGEs carrying antibiotic resistance genes [24]. Lp is an intracellular pathogen responsible for community-acquired severe pneumonia [25], where recombination and HGT drive the emergence of epidemic clones [26], but is not usually antibiotic resistant. Ab and Lp are therefore complementary models for revealing commonalities, and specificities, in the evolution of natural transformation. Here, we obtained 2 very large sets of genomes and transformation rates to characterize the distribution and evolutionary pace of these rates and to test if the trait is under selection. We use them to assess the hypothesis that genetic conflicts caused by MGEs contribute to explain variations in transformation rates.

## Results

### Variable transformation rates across *Acinetobacter* and *Legionella* strains

We analysed 496 draft genomes of Ab and 786 of Lp. The species pangenomes included 31,103 (Ab) and 11,932 (Lp) gene families. Ab genomes had on average 3,598 genes with a coefficient of variation of 5.0%, whereas genomes of Lp had on average 3,091 genes with a coefficient of variation of 3.3%. We used the gene families present in more than 95% of the genomes (persistent gene families: 2,325 in Lp and 2,629 in Ab) to build 2 types of recombination-aware phylogenetic trees for each species by maximum likelihood (see Methods, S1 Data). The phylogenetic reconstructions and most analyses in this study were done on 3 pairs of phylogenetic trees (ignoring recombination, using random positioning of genes, and using the most likely organization). The qualitative results of all major analyses were similar in all the cases. Hence, only the recombination-aware method using the consensus genome organization is presented in the text. The phylogenetic trees of the 2 species have approximately similar average root-to-tip distances (Ab: 0.031 subst$^{-1}$; Lp: 0.047 subst$^{-1}$), but the Lp tree has many more small terminal branches than Ab. Altogether, the Ab data set is more variable in terms of gene repertoires, but the species are of comparable age (distance to the species' last common ancestor).

We used a luminescence-based assay to quantify the ability of each strain to acquire DNA by natural transformation (see Methods). Bacteria are provided with a linear transforming DNA consisting of the *Nluc* gene flanked by sequences homologous to the chromosome. Integration by natural transformation results in Nluc expression which is detected following addition of furimazine. Transformation assays were performed multiple times and revealed good concordance (S7 Data). Strains known to be non-transformable (Ab Δ*comEC*, Lp Lens) were used to define the minimal detection limit of the method (S2 Data and S2 Fig), which is of the same magnitude in the 2 species: 300 for Lp and 400 for Ab. The distributions of the average values of transformation extend below the detection limit of transformation and have a long tail of higher values spanning several orders of magnitude (Fig 1B). As a result, 52% of the strains were deemed transformable in Lp and 64% in Ab. The range of transformation rates is higher in Ab (6 orders of magnitude) than in Lp (4). This might be due to the more diversified set of strains Ab collection, notably encompassing environmental strains (absent in the Lp data set). Accordingly, Ab environmental strains were associated with greater transformation rates than the clinical ones (phyloglm T2, $p = 1.08 \times 10^{-4}$).

Variations in transformation rates could be caused by differences between the focal strains and the template sequence used to produce linear transforming DNA (reference Ab A118 and Lp Paris strains). Instead, we found that transformation rates are highly variable across the phylogenetic trees (Fig 1A). The rooted recombination-free phylogeny of Lp is divided into 2 large phylogroups. One of them composed of 78 strains was more often associated with non-transformability than the other strains of Lp ($\chi^2 = 41.22$, $p = 1.36 \times 10^{-10}$). This phylogroup is considered to belong to the subspecies *raphaeli* [27]. In Ab, the variations in transformation rates did not correlate with the strains' phylogenetic distance (Ab: Spearman correlation $\rho = 0.0034$, $p = 0.22$) even between closely related strains at a patristic distance of less than 0.02 nucleotide substitutions per site (Ab: $\rho = 0.029$, $p = 0.18$) (S3 Fig). In Lp, the correlation was significant but its effect size as measured by the coefficient of association was extremely low in both cases (all patristic distances: $\rho = -0.046$, $p < 2.2 \times 10^{-16}$; patristic distance less than 0.02 nucleotide substitutions/site: $\rho = -0.034$, $p < 2.2 \times 10^{-16}$) (S3 Fig). The phylogenetic distance between the focal and reference strains are not correlated to the differences in transformation rates for the most distant strains (patristic distances larger than 0.04 nucleotide substitutions per site) (Ab: $\rho = -0.011$, $p = 0.81$; Lp: $\rho = 0.029$, $p = 0.80$, S4 Fig). When focusing on the actual homology arms, all strains presented homology arms with more than 99.9% identity with the donor plasmid in Ab and in Lp. In Ab, transformation rate was not correlated with the divergence observed on the homology arms (S5 Fig, $\rho = -0.026$, $p = 0.54$). In Lp, there was a significant correlation between transformation rate and divergence at the homology arms (S5 Fig, $\rho = 0.50$, $p < 2.2 \times 10^{-16}$) even when focusing on the closest strains to the plasmid donor strain ($\rho = 0.48$, $p < 2.2 \times 10^{-16}$). This effect size is not negligible but the important variance we observed in the transformation rates for any percentage of identity of homology arms prevented us from finding a form of normalization that would correct for this association. However, transformation rates are known to be robust to sequence divergence of 1% to 5% [28,29] which is much higher than what we observed here (>99.9%), suggesting that the correlation observed in Lp between sequence divergence and transformation rates might be indirect. In conclusion, transformation rates exhibit high variability across the species which is not explained by the phylogenetic distance between the donor and the recipient bacteria. The sequence divergence between the recipient chromosome and the homology arms of their transforming plasmid is not affecting the patterns of transformation rates in Ab but could explain a part of the variation in Lp.

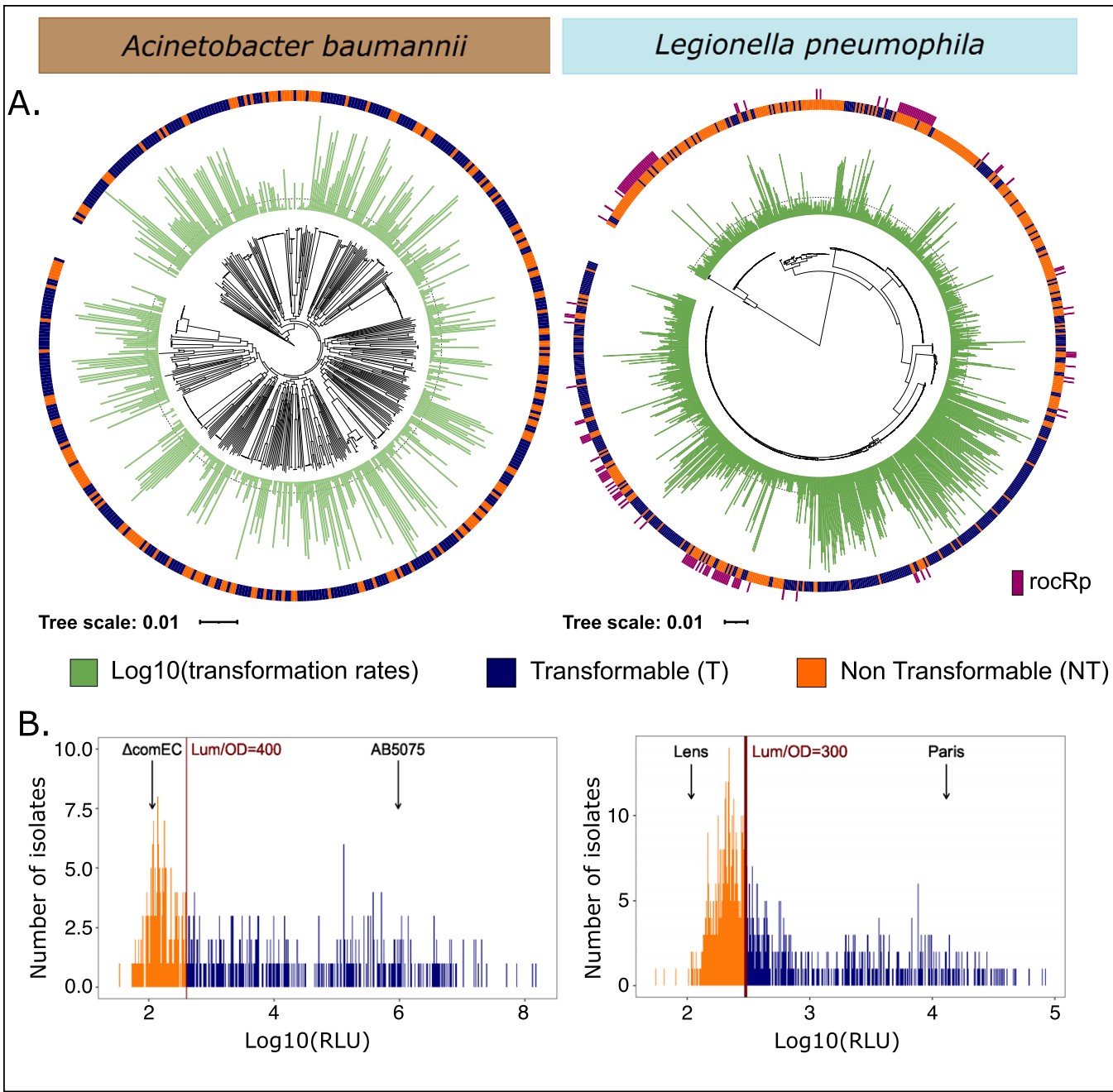

**Fig 1. Distribution of the transformation phenotype.** (A) Distribution of the log10-transformed average transformation rates and of the binary trait across the recombination-free rooted phylogenetic trees of *A. baumannii* (left) and *L. pneumophila* (right). sncRNA rocRp presence (dark pink) distribution is also represented across *L. pneumophila* phylogenetic tree. The dashed line in the log10-transformed average transformation rates distribution corresponds to the transformation rate threshold that separates transformable from non-transformable strains. Tree scale is in substitution per site. (B) Distribution of the log10-transformed average transformation rates in *A. baumannii* (left) and *L. pneumophila* (right). The red vertical line stands for the threshold between transformable and non-transformable strains. ΔcomEC Ab strain and Lp Lens strain are known non-transformable strains. AB5075 Ab strain and Lp Paris strain are among the most transformable known strains. The data underlying this figure can be found in S8 Data.

## Transformation rates evolve according to a jump process

The analysis of transformation frequencies in the light of the phylogeny of the species revealed high variability. We wished to study this variability and how it unfolds in time to understand the evolutionary processes at play. The simplest model of evolution is when the trait follows a Brownian motion(BM)-like process, i.e., when it endures incremental changes drawn from a random distribution with zero mean and finite constant variance. We used the classical Pagel's λ and Blomberg's K indices to assess this model [30–32], but they provided inconsistent results (S3 Data). According to Pagel's λ, the trait might evolve according to a Brownian motion model (S6 Fig), but Blomberg's K could not rule out the hypothesis that the trait was randomly distributed across the phylogeny. To understand these contradictory results, we used the Fritz and Purvis's D statistic to model the binary trait and quantify the strength of the phylogenetic signal on it [33]. This statistic indicated that closely related strains had a less similar phenotype than expected under a Brownian model but more than if it was random (Lp: D = 0.52; Ab: D = 0.58). This means that the Brownian model alone is not sufficient to explain the evolutionary dynamics of natural transformation.

The wide variation of the transformation trait across the phylogenetic trees led us to consider complex models that extend the classical Brownian motion model by accounting for jump processes, i.e., that account for evolutionary processes where the trait may occasionally change abruptly [34]. We assessed how 9 models with diverse evolutionary dynamics fitted the evolution of the transformation rates (log transformed, S4 Data). Six of them use Lévy processes with 2 components: a Brownian motion and a pure jump process (S6 Fig). In a jump process, the trait has abrupt changes that are punctual or occur over a short period of time. As expected, the Brownian motion model did not fit the data very well (Ab: $AIC_c$ = 3,521; Lp: $AIC_c$ = 2,198). The best fitting models in both species were all jump processes (Jump Node, Variance Gamma, Brownian Model + Normal Inverse Gaussian, and Brownian Model + Variance Gamma, S4 Data), which had all approximately similar fit with the data (similar $AIC_c$) because they only differ in the type of random distribution defining the frequency and the amplitude of the jump (S6 Fig). The good fit of these models suggests that natural transformation evolves as the result of 2 processes. First, a gradual process of divergence that is captured by the Brownian model and could result from mutations of small effect on the trait, e.g., point mutations with effect on gene expression of the machinery of transformation. Second, the Brownian process is complemented by frequent sudden changes that separate periods of relative stability and which are better captured by the jump process. Such sudden changes could be explained by the acquisition or loss of genetic determinants of transformation, i.e., they are compatible with the well-known impact of horizontal gene transfer in bacterial evolution which can lead to large sudden phenotypic changes [35].

## Loss of transformation is counter-selected

The previous results suggest sudden transition between transformability and non-transformability. If transitions result from intragenomic conflicts, the latter must result from the deleterious impact of the loss of transformation for bacterial fitness. To test this hypothesis, we inferred the ancestral states of transformation in the phylogenetic tree and searched for recent transitions, i.e., those occurring in terminal branches. More than a fifth of the terminal branches had a phenotype transition (Lp: 20%; Ab: 20%), and there was a very large excess of transitions to non-transformability relative to those towards transformability (Lp: 79% of all events, $\chi^2$, $p < 2.2 \times 10^{-16}$; Ab: 81%, $\chi^2$, $p < 2.2 \times 10^{-16}$). If the process was at equilibrium, there should be an equal number of transitions in both directions. Excess of transitions towards one of the states is usually a sign of purifying selection, i.e., a process where genetic

changes enrich populations in a state (here, non-transformability) that is deleterious and therefore gradually purged by natural selection [36].

If the loss of transformation is deleterious for bacteria, then it should impact the evolutionary trajectories of bacterial lineages. Notably, if lineages of non-transformable strains are less fit then they are expected to be shorter on average than the others because the lineage is more rapidly lost by natural selection (or reverts to the other phenotype). We took all terminal branches where there is no change in the phenotype and analysed their lengths. This revealed much shorter branches in non-transformable strains in Ab (90× shorter, Wilcoxon, $p = 1.14 \times 10^{-09}$) but slightly longer ones in Lp (0.4× shorter, Wilcoxon, $p = 0.03$). Again, this suggests that non-transformable strains, at least in Ab, tend to be removed from the population by natural selection. It should be noted here that the reasons for the counter-selection of non-transformable strains can be varied and depend on the reason of existence of natural transformation. If transformation is adaptive mostly because it removes MGEs from genomes, then the cost of its loss is the accumulation of costly MGEs in lineages (as shown below).

## Transformation impacts recombination rates and genetic linkage

Some of the proposed causes of selection for transformability, e.g., DNA as a source of nutrient or for repair, cannot be tested with our data. But the hypothesis that natural transformation facilitates adaptation by favoring allelic exchanges can be tested. We identified recombination tracts in the terminal branches of the species trees which covered 2,628 persistent gene families in Ab and 2,325 in Lp (see Methods). When considering the phylogeny, transformable strains exhibited a higher recombination rate than the non-transformable ones (phyloglm T17, Lp: $p = 1.29 \times 10^{-15}$; Ab: $p = 1.36 \times 10^{-3}$). Hence, we can identify the expected association between recombination rates and transformation. To better understand the relation between recombination and transformation, we analysed strains whose transformation phenotype changed recently. We also observed greater cumulated lengths of recombination tracts when the strain became transformable in the extant branch than when it became non-transformable, but the result was significant only in Ab (Fig 2A and S5 Data). This suggests that the gain of transformation is responsible for an increase in recombination rates but only in Ab.

Since transformation favors allelic exchanges, it could have an important impact in breaking genetic linkage. This hypothesis can be tested by analyzing the patterns of linkage disequilibrium. We calculated the squared correlation ($r^2$) between bi-allelic values at 2 loci in windows of 500 nt along the genome of transformable and non-transformable strains. The difference between the 2 ($\Delta r^2$) was significant (Lp: average 0.018, Wilcoxon, $p < 2.2 \times 10^{-16}$, Ab: average 0.0092, Wilcoxon, $p < 2.2 \times 10^{-16}$) (Figs 2B and S4), indicating higher correlation, hence higher linkage disequilibrium, in non-transformable strains. This agrees with the above hypothesis and suggests that transformation may break deleterious allele combinations or create novel adaptive ones.

## Transformation rate variations rarely depend on its molecular pathway

To understand the causes of the loss of transformation, we focused initially on the most obvious candidate genes: the ones directly involved in the molecular process (Fig 3). They are the most susceptible to influence the transformation phenotype. We made a survey of the literature to list genes associated with transformation and searched for their presence in every genome (S11 and S12 Data). Very few of these genes are ever missing in Lp, suggesting that inactivation of the transformation genes is rarely the cause of variations in this species. The notable exception is the gene *gspH/fimT* which was missing in some Lp strains, most often in the non-transformable ones. It was previously shown that this gene is dispensable for

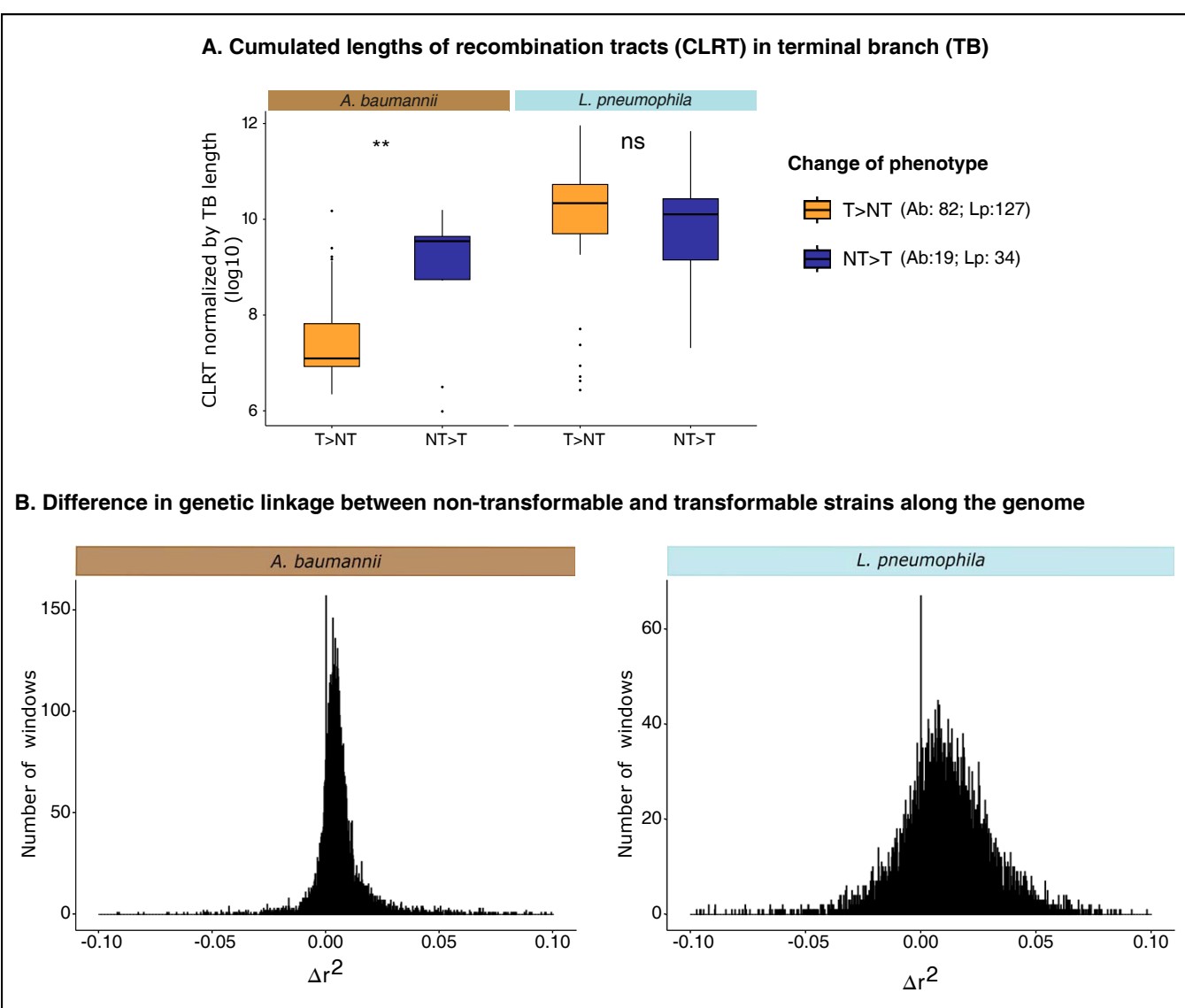

**Fig 2. Association of the transformation phenotype on recombination.** (A) Distribution of CLRTs (Log10-transformed) in TBs. The CLRT was normalized by the length of the terminal branches in relation to the inference of changes in the phenotype for these groups to be comparable. Wilcoxon test, *: $p < 0.05$, **: $p < 0.01$, ***: $p < 0.001$, ****: $p < 0.0001$. (B) Distribution of $\Delta r^2$ values in the $[-0.1;0.1]$ interval computed in 500 nt screened windows where $r^2$ was not null in both T and NT populations. The full span of the distributions is presented in S7 Fig. $\Delta r^2$ was calculated as $r^2_{mean}(NT)-r^2_{mean}(T)$. The data underlying Fig 2A can be found in S9 Data and Fig 2B in S10 Data. CLRT, cumulated length of recombination tract; TB, terminal branch.

transformation in Lp [37], which is consistent with the observation that it is lacking in some transformable strains. Our results suggest that its loss may be associated with lower frequency of transformation (on average 0.8-fold less transformable) that in some cases drops below the detection level. The gene *pilQ* was interrupted by a phage in 1 Ab strain and pseudogenized in 3 Lp (all 4 strains are non-transformable). In Ab some genes for minor pilins could not be identified in a few genomes of both transformable and non-transformable strains (PilEVX). Knock-out of minor pilins was enough to block transformation in Ab W068 [38]. Their absence in transformable strains may be explained by the minor pilins having a more diverging sequence than our search constraints would allow. Pilins evolve quickly by point mutation, HGT and duplication processes, complicating their detection in draft genomes. More

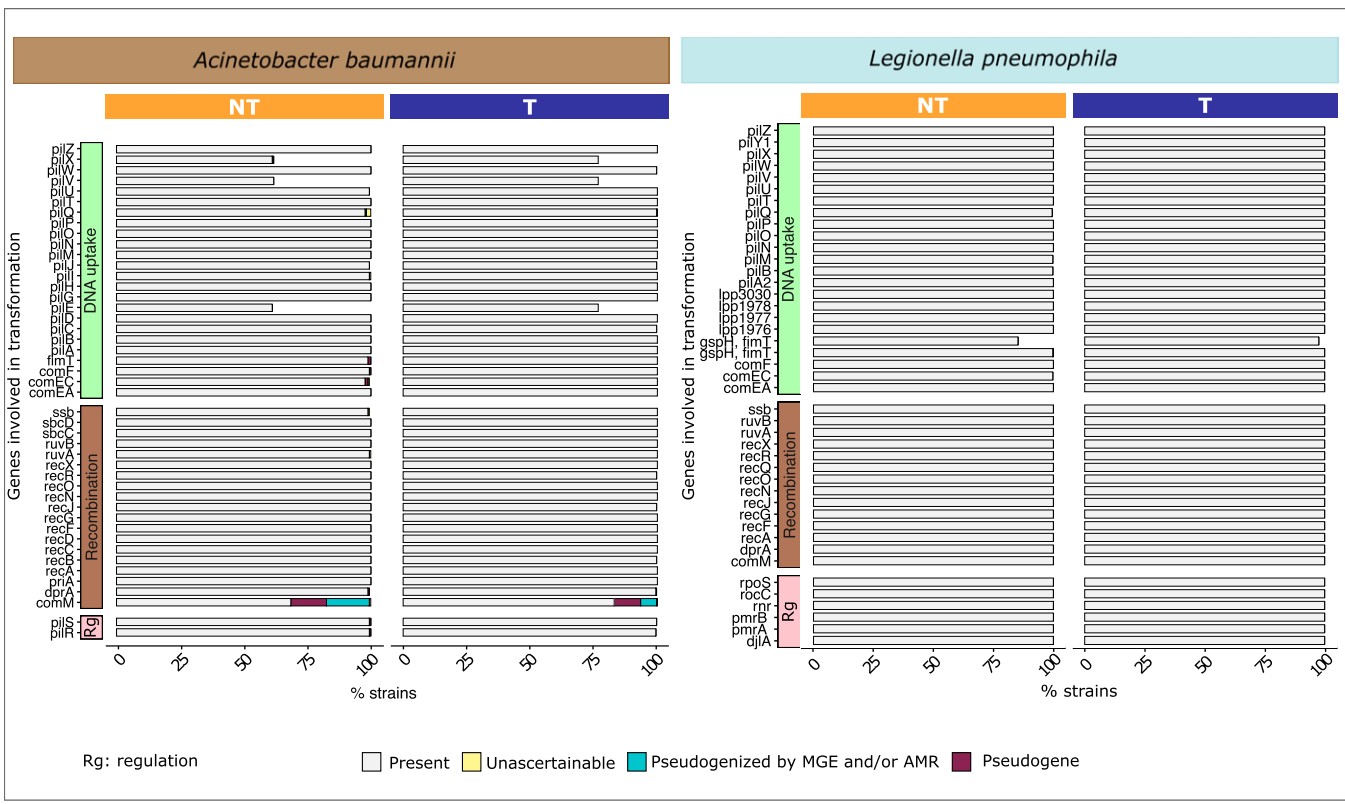

**Fig 3. Presence of genes involved in natural transformation in *A. baumannii* (left) and *L. pneumophila* (right).** The genes were divided regarding their function: DNA uptake, recombination, and regulation (Rg) of transformation. The data underlying this figure can be found in S12 Data.

importantly, the gene *comM*, encoding a helicase involved in the homologous recombination of transforming DNA into the chromosome, was often pseudogenized. This was significantly more frequent in non-transformable than in transformable strains (phyloglm T20, $p = 9.45 \times 10^{-9}$) (Fig 3) and in clinical strains relative to environmental ones (phyloglm T21, $p = 6.47 \times 10^{-8}$). The inactivation of this gene was most often caused by its interruption in the MgCh domain by the integration of MGEs. This has been described in clinical strains, where AbaR and AbGRI resistance islands integrate in this region. But contrary to previous analyses our data set has only a small number of clinical strains. Our detailed analysis of this locus revealed AbaR islands in more than half of *comM* inactivations. In the other cases, we found in these regions MGEs with a plethora of insertion sequences, integrons, and defense systems (S8 Fig). For example, the gene was interrupted by a locus encoding CBASS (40% of the times) and Zorya (6%). The latter interruption was only observed in environmental strains. This is consistent with the existence of genetic conflicts between MGEs and the host regarding natural transformation beyond the selection for the spread of antibiotic resistance genes in clinical strains.

Interrupted transformation genes might be recovered by recombination and allow re-acquisition of the transformation phenotype. To test this possibility, we analysed the patterns of recombination on the locus in the 386 genomes of Ab with a complete *comM*. We found that *comM* was covered by a recent recombination tract in 35 samples. This raises the possibility that inactivated *comM* genes may become functional again by recombination with homologous DNA arriving from other strains. This process is easier to achieve for genes like *comM* whose inactivation does not completely block recombination. Yet, recombination salvaging

*comM* does not happen more often than expected by chance ($\chi^2$, $p = 0.72$). With the exception of *comM* in Ab, variations in transformation rates were rarely explained by the inactivation of transformation-related genes.

Ab can be capsulated (unlike Lp) and it has been recently suggested that its capsule may affect transformation rates [39]. We searched for capsule loci in all Ab strains using *Kaptive* [40]. All genomes presented one presumably complete locus suggesting that capsule inactivation was not causing the emergence of transformable strains. Future work will be needed to test if changes in the expression of the capsule could contribute to the variation in transformation rates in Ab.

## MGEs shape transformation rate variations

To identify the genetic determinants responsible for shifts between transformability and non-transformability, we performed a unitig-based Genome-Wide Association Study ($GWAS_U$) on the binary transformation phenotype ($GWAS_U^{bin}$). We used a linear mixed model (LMM) to correct for population structure using the recombination-aware phylogenetic trees. We applied a Benjamini–Hochberg adjustment for multiple tests at 0.05 (S9 Fig), because Bonferroni corrections are known to be excessively conservative [41]. Indeed the latter led to loss of hits that were experimentally verified to affect transformation rates (ComM, see Methods). The relevant genetic variants were mapped to the gene families. In Lp, 378 gene families including 1 sncRNA gene were associated with the inhibition of transformation and 271 were associated with increased transformability (Fig 4A and S6 Data). In Ab, 836 gene families were associated with the inhibition of transformation and 426 with increased transformability (Fig 4A and S6 Data). In both species, it is thus less than 5% of the pangenome that could explain the non-transformable phenotype. Given the linkage between genes, this proportion might be overestimated. These gene families did not include any capsule loci genes. Among the genes directly involved in natural transformation only unitigs mapping *comEA*, *rnr*, *recQ* in Lp and *pilP*, *pilT* in Ab were associated with non-transformability. Since these genes are part of the persistent genome, this suggests that some variants may lower transformation rates. Only the unitigs mapping *comM* were positively associated with transformability, a consequence of the abovementioned frequent inactivation of this gene. This confirms the impact of *comM* inactivations on transformation and suggests that natural sequence variants of the other genes may affect transformation rates. Overall, the genes directly implicated in the pathway of natural transformation are a very small fraction of all genes identified by the $GWAS_U^{bin}$.

In both species, recognizable MGEs are important genetic determinants of transformation inhibition. We classed them in ICEs, plasmids, phages, and transposable elements (Fig 4B). Beyond the genes that were individually significantly associated with low transformation, many of the other MGE genes were collectively negatively associated with the loss of transformation, even when each individual effect was not significant (Fig 4B). For example, many Insertion Sequences, sometimes part of larger elements like plasmids and ICEs, are among transformation-inhibiting candidates in both Ab and Lp (Fig 4B).

Several prophage genes are negatively associated with transformation in Ab. Some of these functions can also be found in other elements, but others are very specific to phages as revealed by the analysis of their viral quotients (see Methods). Out of the 836 Ab transformation-inhibiting gene families from $GWAS_U^{bin}$, 24 matched proteins with very high viral quotient (higher than 0.9, see Methods). Among them, the most significantly transformation-inhibiting gene seems to be part of a prophage anti-defense system. We predicted its protein structure with AlphaFold and aligned it against a large data set of protein structures using Foldseek. This analysis revealed structural homology to DarA, a protein that is involved in capsid

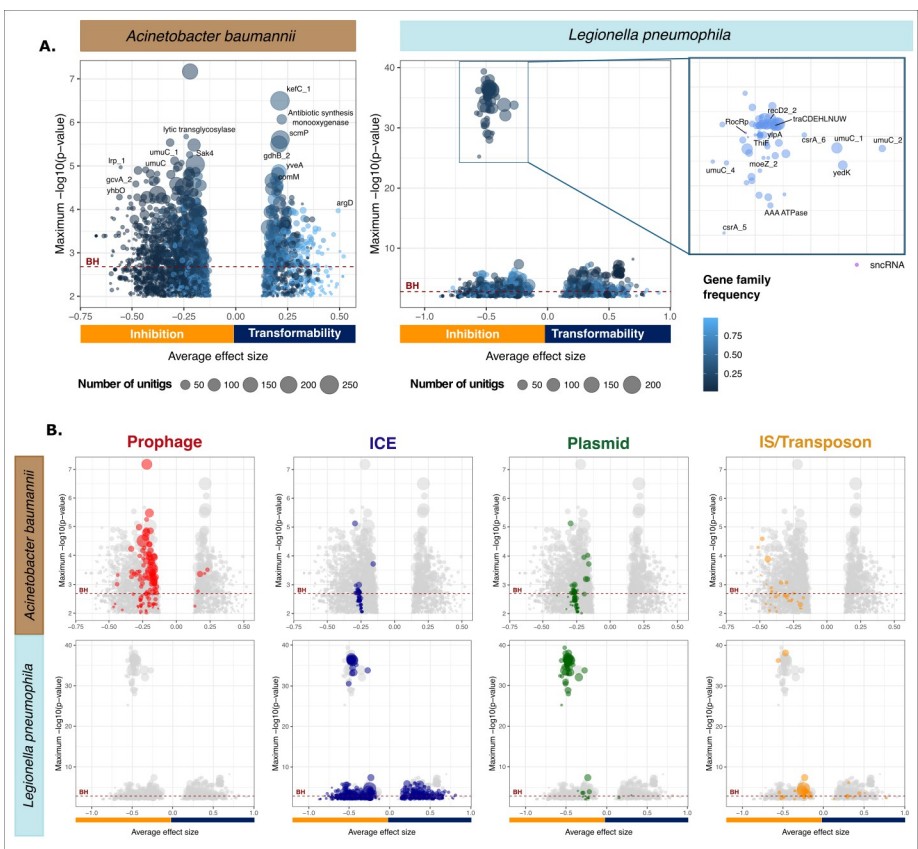

**Fig 4. Volcano plots showing average effect sizes and significance of the association of the gene families with the transformation phenotype according to $GWAS_U^{bin}$ in *A. baumannii* and *L. pneumophila*.** Each circle stands for a gene family. The size of the circle depends on the number of unitigs that mapped the gene in all the samples. The value on the x-axis corresponds to the average effect size of all the unitigs mapping the gene. The y-axis indicates how significant this effect can be by representing the maximal -log10-transformed *p*-value adjusted for population structure (lrt-pvalue) of all the unitigs of this gene. The BH threshold (dark red line) of 0.05 is set at the lrt-pvalue that once corrected by BH is equal to 0.05. Significantly associated gene families are above the Benjamini–Hochberg (BH) threshold (red dashed line). The lower graphs (B) are similar to the ones on top (A), but gene families were colored in respect to their MGE. Only gene families whose annotation was known and that had either a strong effect size or were very significant were labeled for readability. The whole set of gene families, their *p*-values, effect size, and frequency is listed in S6 Data. The data underlying this figure can be found in S6 Data. GWAS, genome-wide association study; MGE, mobile genetic element.

morphogenesis and is a component of an anti-restriction system (found in *A. baumannii* ACICU, S10 Fig) [42]. However, the homology is restricted to the N-terminal domain and this gene stands alone in the GWAS (the Dar system being composed of many proteins). Further work will be necessary to disentangle the function of this protein. In brief, the abundance of prophage genes in the GWAS suggests these elements have an important role in the inhibition of transformation in Ab.

We could not identify intact prophages in the Lp strains. Instead, the transformation-inhibiting candidates in this species were often associated with functions found on plasmids, including conjugation and the plasmid-encoded sncRNA RocRp (Fig 4). A little more than 15% of the Lp transformation-inhibiting genes identified in $GWAS_U^{bin}$ (62/378) were carried by plasmids. It was previously shown that RocRp is plasmid-encoded and inhibits transformation [21]. Our results show that similar plasmids lacking RocRp do exist (S11 Fig). Importantly,

none of the RocRp copies could be identified in the chromosomal contigs. RocRp plasmids also lacked serine/tyrosine recombinases as well as *dif* sites or XerL homologs that could lead to Xer-mediated integration in the chromosome. To confirm that contig assignment was not affecting our results, we searched 113 complete Lp genomes from RefSeq where all 5 occurrences of RocRp were also in plasmids that had high similarity in gene repertoires (S11 Fig). Of note, some plasmids lacking RocRp had moderately high wGRR (>0.6) with the RocRp plasmids. Overall, 32% of the non-transformable strains encoded RocRp plasmids. This raised the following question: Are the plasmid-associated genes in the $GWAS_U^{bin}$ found because of their genetic linkage with rocRp or because they are independently associated with the inhibition of transformation? To answer this question, we built an LMM where the presence of rocRp in the strain is a covariate of $GWAS_U^{bin}$ ($GWAS_U^{bin}$-cov). This model retained some significant associations between plasmid-associated genes and transformation in Lp (BH adjusted *p*-value <0.05, S12 Fig), but the overall *p*-values of these genes were much smaller. In fact, most of these genes were conjugation genes that also exist in ICEs suggesting their association might be retained because of some other transformation determinants that ICEs carry. Hence, the negative association between plasmids and transformation is largely driven by genetic linkage with the transformation-inhibiting sncRNA rocRp, albeit some other genes present in certain ICEs may be good candidates for secondary modulators of transformation rates.

Given the importance of MGEs in shaping transformation rates, anti-MGE defense systems could directly lower transformation rates by targeting incoming DNA [43] or indirectly increase transformation rates by preventing the acquisition of MGEs encoding transformation-inhibiting genes. We separated innate (e.g., restriction-modification systems) from adaptive (CRISPR-Cas) defense systems because the former may block MGEs and bacterial DNA arriving by transformation, whereas the latter are only expected to block MGEs (since they provide a specific defense). We detected a negative association between the number of putatively innate defense systems of a strain and its transformability in Lp and in Ab (phyloglm T12, Lp: $p = 1.03 \times 10^{-13}$; Ab: $p = 3.61 \times 10^{-9}$) and more specifically when the defense system was a restriction-modification system (phyloglm T13, Lp: $p = 4.34 \times 10^{-8}$; Ab: $p = 1.30 \times 10^{-6}$). Of note, these unitigs did not map the R-M systems found in Vesel and colleagues [43], showing that other R-M systems also contribute to diminish transformation rates. In contrast, there was a positive association between the number of CRISPR-Cas systems carried by a strain and its transformability in both species (phyloglm T14, Lp: $p = 7.81 \times 10^{-9}$; Ab: $p = 0.018$). These results suggest that defense systems impact transformation rates in ways depending on their ability to specifically target MGEs.

Since both MGEs and innate defense systems, especially restriction-modification systems, are associated with lower transformation rates, we quantified how much variation in transformation rates could be explained by each of them. We performed a phylogenetic logistic regression expressing the binary transformation phenotype as a function of the number of mobile genetic elements inhibiting transformation (phages in Ab and conjugative systems in Lp) and of the number of restriction-modification system a strain carries. As expected, both were significatively and negatively associated with transformation, but phages in Ab (phyloglm T22, effect = −0.97 [−1.038; −0.905], $p = 4.3 \times 10^{-3}$) and conjugative systems in Lp (phyloglm T22, effect = −0.76 [−0.779; −0.735], $p = 1.4 \times 10^{-4}$) had a stronger inhibitory effect than restriction modification systems (phyloglm T22, Ab: effect = −0.61 [−0.634; −0.590], $p = 3.55 \times 10^{-8}$; Lp: effect = −0.41 [−0.441; −0.398], $p = 1.09 \times 10^{-9}$). This suggests that variations in transformation rates are more impacted by MGEs inhibitory effects than by the action of restriction-modification systems.

## Transformation is associated with the loss of MGEs

The chromosome curing model suggests that genetic conflict between MGEs and the host arise because transformation cleans the bacterial chromosome from its MGEs. One would thus expect to find fewer MGEs in transformable strains than in the others. We analyzed the frequency of MGEs in relation to the transformation phenotype and the phylogenetic structure using a phylogenetic logistic regression [44]. Even though some prophages are negatively associated with transformation (see above), the number of prophages in Ab does not depend on the strain transformation phenotype ($p > 0.05$; no prophages in Lp). When compared to the others, the chromosomes of transformable strains carry fewer Insertion Sequences (Fig 5; phyloglm T19, Lp: $p = 5.97 \times 10^{-5}$; Ab: $p = 0.0016$) and fewer conjugative systems (phyloglm T7, Lp: $p < 2.2 \times 10^{-16}$; Ab: $p = 1.21 \times 10^{-7}$). Of note, conjugative plasmids were significantly rarer in transformable strains in both species (phyloglm T9, Lp: $p < 2.2 \times 10^{-16}$; Ab: $p = 1.38 \times 10^{-8}$). Hence, many types of MGEs are less abundant in transformable strains.

If recombination cures the chromosome from MGEs, one expects to find an excess of homologous recombination targeting persistent genes that flank MGEs. For each persistent gene, we used the number of times it was covered by a recombination tract in our collection as a proxy for its recombination rate. Based on the MGE flanking persistent genes previously identified in the whole collection, we were able to evaluate for each persistent gene if it had a MGE in its direct neighborhood in at least 1 genome. We observed that among persistent genes, those in the direct neighborhood of an MGE had higher recombination rates in Ab: conjugative systems (Ab: Wilcoxon, $p < 2.2 \times 10^{-16}$), and insertion sequences (Ab: Wilcoxon, $p < 2.2 \times 10^{-16}$), but not for phages (Ab: Wilcoxon, $p = 0.08$). In Lp the association was significant when the MGE was an insertion sequence (Lp: Wilcoxon, $p = 1.7 \times 10^{-5}$). Interestingly, recombination also targets at high frequency the persistent genes neighboring defense systems (Ab: Wilcoxon, $p < 2.2 \times 10^{-16}$), which fits previous observations that these systems evolve rapidly and are often within MGEs [45]. In conclusion, core genes flanking MGEs show an excess of recombination tracts, as expected if transformation removes neighboring MGEs by

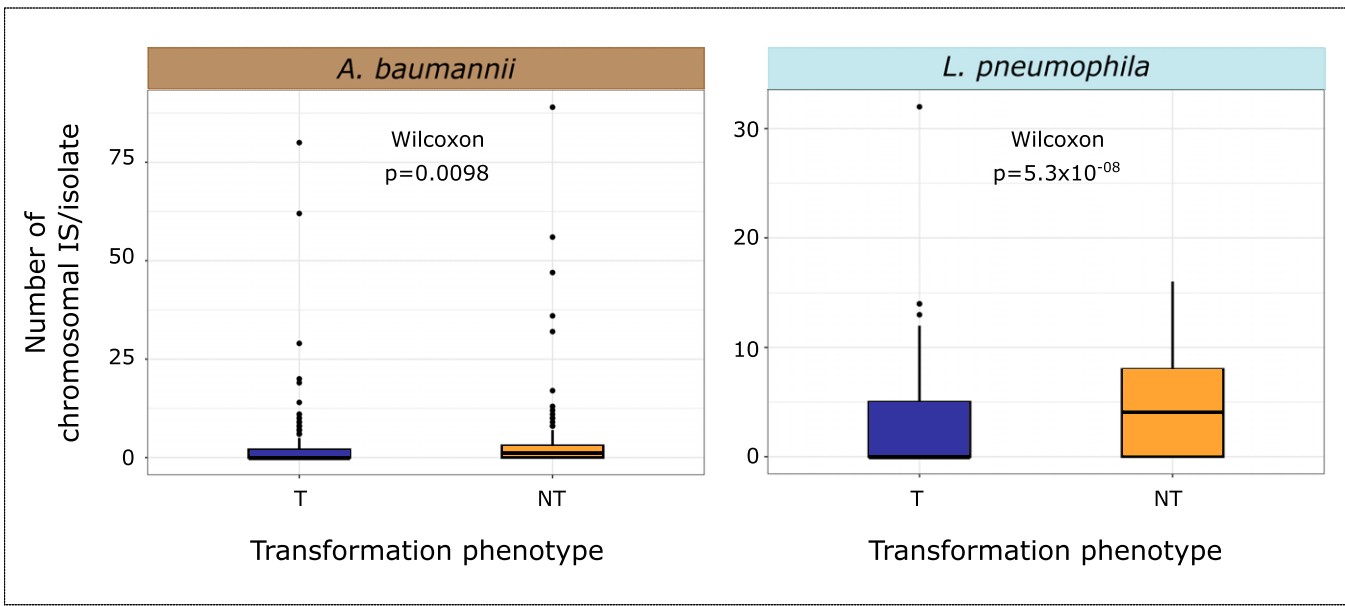

**Fig 5. Distribution of the number of Insertion Sequences in the bacterial chromosome per isolate in transformable and non-transformable strains in *A. baumannii* (left) and *L. pneumophila* (right).** The data underlying this figure can be found in S14 Data.

recombination. It is worth noting that such events of recombination may also occur by incoming DNA arising directly from the action of MGEs, e.g., by HFR-like conjugation [46] or lateral transduction [47]. These removals may become fixed in the population when they increase bacterial fitness.

## Discussion

Ab and Lp are 2 bacterial species with very different lifestyles. Notably, Lp is intracellular when Ab is not. Lp collection included only clinical isolates and is thus much less diverse than the Ab collection which included mainly environmental strains in addition to some clinical isolates. This was visible on the phylogenetic trees but also in their gene repertoires which were broader in Ab compared to Lp. Both presented important variations in their transformation rates across the phylogenetic tree but only Lp showed some association between the transformation rates and its population structure. This might find its root in the division of the collection in 2 clades with 1 corresponding to the *raphaeli* subspecies of Lp. This could also result from higher transformation rates between more similar recombination arms. However, such an association was previously shown not to operate when sequence similarity is very high [28,29]. Its constancy in the range 0% to 5% divergence is the basis of many previous studies [48–50]. The data set of Ab has a mixture of clinical and environmental strains and this allowed us to discover that environmental strains were more transformable than the clinical ones. This could explain the greater amplitude in transformation rates we observed in Ab collection compared to the fully clinical Lp collection. Ab exhibited a specific and frequent mechanism of inhibition of natural transformation. Indeed, *comM* was frequently inactivated in non-transformable Ab strains. The inactivation of *comM* that is responsible for a reduction in the transformation rates does not exist in Lp, possibly because the latter lacks the Tn7-like element that targets the *comM* gene in Ab. However, the absence of Tn7-like elements does not fully explain why *comM* interruption is observed in Ab and not in Lp since some *comM* interruptions in Ab are not caused by Tn7-like elements. In addition to AbaR, *comM* is also interrupted by other IS and defense systems notably in environmental strains. One possible explanation might lie in the intracellular lifestyle of Lp where recombination with distantly related lineages might be lower than for Ab, which would render the ComM function less critical for solving heterologous recombination and complete natural transformation.

We found wide within-species variations in transformation rates under similar growth conditions, with 64% of Ab and 52% of Lp being transformable beyond the detection limit of the method. Variations in transformation rates were previously shown in smaller samples of other species. In *A. actinomycetemcomitans*, only 26% of strains are competent for transformation [51], and in *S. pneumoniae* and *H. influenzae* [17,19] around two thirds of the strains are transformable. The ranges of variation in transformation observed in this study (4 orders of magnitude in Lp and 6 in Ab) are close to the ones of *H. influenzae* (6) and *S. pneumoniae* (4). They are probably underestimates because transformation rates may vary between environments (for which there is no data available). We found very little association between transformation rates and population structure, as previously observed in *S. pneumoniae* [17], but contrary to observations in *P. stutzeri* [51]. Instead of transformation rates following species phylogroups they are associated with the patterns of distribution of MGEs.

Allelic recombination decreases genetic linkage thereby rendering natural selection more efficient. Recombination tends to facilitate the fixation of adaptive mutations and alleviates the cost of deleterious ones [52], especially if it is fitness-associated [53]. It could thus be one of the key advantages of transformation if this process is frequent when bacterial populations are diverse, i.e., there are polymorphisms in the population (otherwise recombination does not

break genetic linkage). We found that transformable strains have slightly higher recombination rates than non-transformable ones. This effect is relatively weak suggesting that recombination rates are poor proxies for transformation rates. This could further suggest that selection for allelic exchanges is not a major driver of natural transformation. The low association between transformation and recombination rates may also be caused by the confounding effect of recombination mediated by the transfer of DNA by MGEs, e.g., conjugation or transduction, or by technical difficulties in identifying all recombination tracts. Strains with few MGEs could tend to recombine by natural transformation, whereas those with many MGEs might tend to transform at low rates but still recombine as the result of conjugation or transduction. While natural transformation is expected to result in the recombination of persistent genes across the entire chromosome, MGEs will favor recombination of genes close to the regions where they integrate in the genome [47,54]. This could explain why there is a weak positive association between transformation and recombination rates, but a clearer loss of genetic linkage associated with transformation: for similar rates of recombination, transformation provides DNA covering the core genome more uniformly thereby decreasing genetic linkage more efficiently than MGE-driven recombination that tends to favor the transfer of core genes close to the MGE integration site. To test this hypothesis, it will be necessary to disentangle in the future the contribution of different mechanisms of HGT to homologous recombination.

Even if we cannot rule out that differential expression of transformation-related genes may be responsible for the existence of many non-transformable strains, the simplest mechanistic explanation for their existence is the inactivation of these transformation-related genes by MGEs [13]. One key gene (*comM*) in one of the species (Ab) was often inactivated (22% of the strains) because of AbaR islands in clinical strains and many other diverse MGEs in the others. The extensive collection of Ab environmental strains we have used shows that *comM* is targeted by a much broader range of elements often encoding defense systems. ComM inactivation was also previously described in *A. actinomycetemcomitans* [55] and several *Pasteurellaceae* [56], raising an interesting question: Why would ComM be specifically targeted for inactivation? The function of this protein was recently clarified. It is a helicase that facilitates homologous recombination between heterologous DNA [7]. Its loss reduces transformation by 2 orders of magnitude for the type of DNA tested in our assay (non-homologous segment flanked by homologous regions) [7]. In *V. cholerae* its absence has a very minor role in transformation with identical DNA [57]. This may explain why strains lacking ComM still have measurable rates of natural transformation. The lower recombination with heterologous DNA in *comM* mutants may have a small impact on the host fitness, since DNA uptake and recombination between homologous DNA may still take place. Yet, it may diminish the ability of the bacterium to remove MGEs from the genome (because distant strains with heterologous DNA are more likely to lack the MGE and lead to its deletion from the genome by transformation). Beyond *comM* in Ab, inactivation of competence genes rarely explains the observed variations in transformation rates, possibly because most of these genes also contribute to other important processes, such as DNA repair (recombination), adhesion, and virulence (type IV pilus) [58]. Hence, their inactivation would decrease the host and the MGE fitness.

What explains the high variation in transformation rates across the strains and especially the observed frequent non-transformability? It has been suggested that the persistence of transformable and non-transformable strains is the result of the frequent loss of competence occurring in transformable strains [56]. Non-competent lineages could get a short-term advantage before being selected against in the long term. Many of our results are compatible with the idea that intragenomic conflicts between MGEs and the bacterial host could be responsible for the birth of such non-transformable lineages. We confirmed the negative association between transformation rates and the presence of RocRp-encoding plasmids in Lp [21]

and *comM*-inactivating MGEs in Ab (known for AbaR) [59]. We showed that transformation rates vary following a jump process. Both observations are consistent with a negative impact of MGEs on transformation rates since these elements are frequently gained and lost [60]. Accordingly, we found that MGEs such as ISs, ICEs, plasmids, and phage (depending on the species) are systematically associated with low rates of transformation. It is hard to disentangle which of the MGEs or the low transformation rates is actually the cause or the effect for each MGE: some MGEs may infect the cell and thus block transformation and because transformation is blocked, further MGEs may accumulate in the genome because curing by recombination is stopped. Finally, intragenomic conflicts explain our seemingly contradictory observations that transformation rates are often below the detection limit even though such strains are counter-selected. MGEs often carry traits adaptive in certain ecological contexts and can compensate for a period of time the fitness cost of losing the transformation phenotype. Transformation itself may be costly under certain circumstances and in the short-term non-transformable bacteria might be favored. But in the long term, the inability of these non-transformable bacteria to have the benefits of transformation may result in their counter-selection and the purge of these lineages.

The intragenomic conflict between integrative MGEs and bacterial transformation is a key prediction of the chromosome curing hypothesis [13]. In this model, bacteria use transformation to delete costly MGEs from chromosomes by recombination at flanking persistent genes and the latter strive to counteract this mechanism. This is consistent with most of our observations described above, notably the accumulation of MGEs in non-transformable strains and counter-selection of the latter. It is also consistent with our observation of higher rates of recombination on core genes flanking MGEs. However, this does not explain why conjugative plasmids in Lp are repressing transformation via RocRp. Plasmids cannot be deleted from the chromosome because they are extra-chromosomal. They also usually lack persistent genes and transformation is not expected to delete them. In other bacterial species RocRp can be found in chromosomal islands and in ICEs [61]. Yet, in Lp this gene is exclusively confined to a cluster of closely related, but not identical, plasmids where it has a very strong negative effect on natural transformation. This suggests that intragenomic conflicts between MGEs and bacteria regarding transformation extend to extra-chromosomal elements. We speculate that plasmids might block transformation to prevent the acquisition of incompatible plasmids, exploitative MGEs, or anti-plasmid defense systems. By blocking transformation, RocRp could also contribute to the preservation of chromosomal MGEs in positive epistatic interaction with the plasmid. In addition, plasmids could encode RocRp to avoid recombination with DNA entering by transformation from other partly homologous plasmids that could lead to deletion of non-homologous regions and/or create plasmid instability. In any case, these results strongly suggest that intragenomic conflicts between MGEs and natural transformation are not restricted to the effect of chromosome curing.

The interplay between anti-MGE systems and transformation is complex in the presence of MGE-driven intragenomic conflicts. Defense systems may decrease transformation rates by blocking the entry of exogenous DNA or they may increase them by blocking the acquisition of transformation-inhibiting MGEs. When we excluded the known adaptive defense systems (CRISPR-Cas) we found a negative association between defense systems and transformation rates. Many of these are restriction modification systems, by far the most abundant defense systems of bacteria [62]. Restriction systems were shown to reduce transformation efficiency in specific strains of *P. stutzeri* [63], *H. pylori* [64], *Neisseria meningitidis* [8], *C. jejuni* [65,66], *S. pyogenes* [67], and Ab [43]. CRISPR-Cas are adaptive immune systems and their presence is positively correlated with transformation in both species. The opposed associations of CRISPR-Cas and restriction-modification systems can be explained by the way they work.

CRISPR-Cas systems cannot efficiently protect from transformation because, contrary to R-M, they cannot block generic heterologous DNA but only elements carrying the sequences matching their spacers. Hence, they have probably very little if any role in preventing the acquisition of homologous DNA by transformation. Since they protect the bacterium from the acquisition of MGEs that might block transformation, they have a net positive effect on transformation rates. This means that the impact of defense systems on the variation of transformation rates depends crucially on the specificity and adaptability of their mechanisms of action.

In spite of the differences enumerated above between Ab and Lp, we found striking parallels between these species with different lifestyles and gene repertoires. Both have widely variable transformation rates evolving according to a jump process and decreasing genetic linkage. Non-transformability is scattered across the species trees, seems to be counter-selected, and is associated with the presence of MGEs. This suggests common reasons for the variability of transformation rates. Intragenomic conflicts driven by MGE are compatible with most of the data, suggesting that bacteria–MGE interactions are key drivers of the evolution of natural transformation rates. If intragenomic conflicts seem to have a key role in the evolution of transformation rates, their underlying causes, e.g., chromosome curing of MGEs, are not necessarily the only reasons for the selection for natural transformation. Other putative advantages of transformation, gene transfer, nutrient acquisition, or DNA repair, are lost if the mechanism is inhibited by MGEs. These advantages have the potential to further increase the intragenomic conflict between the bacterium, which benefits from natural transformation, and the MGEs that benefit from blocking it.

## Materials and methods

### Bacterial strains, origin and typing

We analyzed draft assemblies of 830 Lp clinical strains from the Centre National de Référence pour *Legionella* (CNRL) collection and of 510 Ab environmental and clinical strains. We sequence typed Ab collection with *mlst v.2.19.0* (https://github.com/tseemann/mlst). The Lp collection was assembled from clinical isolates collected in France from 2018 to 2020 and was sequence typed by the CNRL. This publication made use of the PubMLST website (https://pubmlst.org/) developed by Keith Jolley [68] at the University of Oxford. We also typed Ab capsules with *Kaptive v.2.0.3* [40,69]. The search for antimicrobial resistance genes was done with ABRicate (https://github.com/tseemann/abricate) which used the following databases ARG-ANNOT [70], CARD [71], MEGARes [72], and VFDB [73].

### Plasmid constructions

Plasmid pJET.Lp-*pilMNOPQ*::*nLuc* was constructed by cloning the *nanoluc* gene (*Nluc*) along with flanking arms of 2,000 bp from the *pilMNOPQ* locus obtained from the genomic DNA of *Legionella pneumophila* strain Paris. Plasmid pJET.Ab-*pilMNOPQ*::*nLuc* was constructed by cloning the nanoluc gene along with flanking arms of 2,000 bp from the *pilMNOPQ* locus obtained from the genomic DNA of *Acinetobacter baumannii* strain A118. The gene *NLuc* is expressed from a synthetic promoter, *Ptac*. This promoter is normally repressed by *lacI*, but since both tested species lack the *lac* operon, the promoter is strong and constitutive.

Plasmids are used uncut because they cannot replicate in the tested species, thus transformants can only be obtained through recombination with the chromosome. The initial step of transformation is fragmentation of the transforming DNA, this has higher chance to occur outside of the cassette when cloned on a plasmid, lowering the chance of cleavage happening within the *NLuc* gene or within the homology arms.

## Natural transformability assay for *L. pneumophila*

*L. pneumophila* strains from frozen stock cultures were thawed, 5 µl of the cells were spotted onto CYE solid medium plates and incubated for 3 days at 37°C. Cells from the plates were subsequently resuspended into 100 µl liquid AYE medium in 96-well plates using a 96-well Scienceware replicator and allowed to grow for 3 days at 37°C in a shaking incubator. Next, 2 µl of this culture was transferred to 100 µl of fresh AYE medium in 96-well plates containing 20 ng/µl of pJET.Lp-*pilMNOPQ*::*nLuc* plasmid DNA and allowed to grow for 3 days at 30°C in a shaking incubator. As this plasmid is non-replicative in *L. pneumophila*, DNA molecules which are internalized undergo a double recombination event allowing the insertion of *nano-luc* gene in the *pilMNOPQ* locus and subsequent expression of the NanoLuc Luciferase enzyme. Subsequently, 80 µl of cells were mixed with 20 µl of the Nano-Glo Luciferase Assay Substrate and Nano-Glo Luciferase Assay Buffer followed by incubation at room temperature for 10 min. The expressed NanoLuc Luciferase enzyme was reported as luminescence units (LUs) on a Promega GloMax Navigator plate reader. The optical density of the cell suspension at 600 nm was detected on a Tecan plate reader. Relative luminescence units (RLUs) were calculated by dividing the luminescence values by the optical density values. RLU values were used as a proxy for transformability of the strains. The assay was repeated independently for each strain in the collection between 3 to 6 times. In Lp, competence occurs at the transition between exponential and stationary [61]. It is possible that some strains display different kinetic of expression. To limit this effect, transformation experiments were conducted with transforming DNA present during the entire growth experiment.

## Natural transformability assay for *A. baumannii*

Natural transformation in *A. baumannii* requires induction by agarose. Agarose soluble extract media (ASEM) was prepared by adding 2 g of Agarose D3 (Euromedex) to a 5 g/L solution of Tryptone media (Bacto). This suspension was vortexed for 5 min and subsequently centrifuged to sediment the insoluble agarose particles. The supernatant was collected and filtered using a 0.22 µm filter. *A. baumannii* strains from frozen stock cultures were thawed, subsequently 2 µl of cells were transferred to LB medium (Lennox formulation) and incubated overnight at 37°C. The following day, 2 µl of cells were transferred to ASEM containing 2 ng/µl of pJET.Ab-*pilMNOPQ*::*nLuc* plasmid DNA using a 96-well Scienceware replicator and allowed to grow overnight at 37°C. As this plasmid is non-replicative in *A. baumannii*, DNA molecules which are internalized undergo a double recombination event allowing the insertion of *nano-luc* gene in the *pilMNOPQ* locus and subsequent expression of the NanoLuc Luciferase enzyme. Subsequently, RLU values were calculated similarly as described above for *L. pneumophila*. The assay was repeated independently for each strain in the collection between 4 to 6 times. In *A. baumannii*, competence is expressed in exponential phase [74]. It is possible that some strains display different kinetic of expression. To limit this effect, transformation experiments were conducted with transforming DNA present during the entire growth experiment.

Recently, an R-M system was discovered that restricts transformation of replicating plasmids but has a lesser effect on transformation of DNA recombining with the chromosome (which is what is tested in our assay) [43]. It should be noted that the transforming DNA used in our assay is extracted from an *E. coli* strain whose methylation system (dam) protects from this *A. baumannii* R-M.

## Validation of the natural transformability assay

We, first, verified that the number of transformants had a linear relationship with the measured luminescence. To do so, we measured the luminescence values of serial dilutions of transformed

bacteria. We isolated 3 transformants of the *A. nosocomialis* M2 strain. They were grown to obtain cultures whose titers were determined (colony-forming unit, CFU, per 100 μl). Then, 10-fold serial dilutions were made and the luminescence of 100 μl suspension was determined. This results in a plot (S13 Fig) in which we can see a linear relationship between CFU count (equivalent to the number of transformants) and luminescence (adjusted $r^2 = 0.99$, $p < 2.2 \times 10^{-16}$). This is true down to $2 \times 10^2$ CFU, which means that we can detect 20 transformants in a well.

We, then, benchmarked our assay by comparing the transformation frequencies determined by selection with a genetic marker and by luminescence values. *A. nosocomialis* M2 strain was transformed with varying amounts of the transforming plasmid, which conveniently carries *Nluc* but also a genetic marker, *kanamycin*. After transformation, the culture was used to determine the transformation frequencies, defined here as the ratio of the determined CFU count of transformants (on selective plates with kanamycin) and the total CFU count (non-selective plate). The luminescence of the same culture was measured. Luminescence measurements and transformation frequencies had a good fit using linear regression (adjusted $r^2 = 0.95$, $p = 1.25 \times 10^{-4}$, S14 Fig).

To ensure the variations in luminescence were not simply the results of variations in expression of the luciferase marker, we transformed a random set of 60 isolates with the *nanoluc* marker (which also confers resistance to kanamycin). Because not all isolates are kanamycin-sensitive, we ended up with 15 transformed isolates. We cultured them and measured the luminescence level. These strains show 100-fold variations in transformation frequencies determined by *nanoluc*, yet quantification of the amount of Nanoluc they produce shows only a 3-fold difference between the minimum and maximum values (S11 Data). Thus, variations in transformation frequencies could not be attributed to variations in expression of *nanoluc* between strains.

To further validate our assay, we tested it with well-documented strains, including mutants whose transformation defects were well known. All these strains and their RLU measurements are to be found in S15 Fig. These results confirm that our system is able to recapitulate published data. Notably, that *comM* inactivation decreases transformation by more than one order of magnitude and that *comEC* inactivation decreases RLU levels to below the threshold we used to class strains as non-transformable. Finally, it shows clear differences between the transformable and non-transformable wild-type strains in terms of RLU.

## Assessment of the transformability phenotype

Since the number of Ab or Lp transformants is linearly correlated with the luminescence signal (S13 Fig), each strain transformation rate was then calculated as the average of the log10-transformed OD-corrected luminescence value on all the replicates. We used the average on all replicates because the assay gave reproducible results between replicates: Spearman correlation coefficients were on average 0.72 in Lp and 0.90 in Ab (S7 Data and S16 Fig). The log10-transformation of the OD-corrected raw luminescence values reduced the skewness of the different replicates and allowed us to compare them. The threshold between transformable and non-transformable strains (Lp: LumR/OD = 300, Ab: LumR/OD = 400) was set based on the maximum of the transformation rates of a non-transformable strain: the Lens strain for Lp (120 replicates) and an engineered ΔcomEC strain for Ab (28 replicates). We chose to use the maximum as a threshold to ensure that any strain we classified as transformable with a transformation rate beyond this threshold could not actually be non-transformable.

## Construction of pangenomes

We removed poor quality draft assemblies with *PanACoTA v.1.2.0* [75] keeping drafts with less than 100 contigs when the sum of the 100 largest contigs was at least 90% of the genome

(L90 $\leq$ 100). We excluded strains that were too similar to already included strains or too distant from the other strains in terms of mash distance to be part of the species (Lp = [$10^{-6}$;0.1], Ab = [$10^{-6}$; 0.06]). After filtering, we were left with 786 Lp isolates and 496 Ab isolates, all listed in S8 Data.

We annotated the draft assemblies (*prokka* from *PanACoTA v.1.2.0* completed with *eggnog-mapper 2.1.9*) and built the pangenome of each collection using single-linkage clustering to form families of proteins with at least 80% identity (using *mmseqs2 v.12-113e3* within *PanA-CoTA v.1.2.0*). We defined the persistent genomes of each species. A pangenome family was considered persistent if at least 95% of the genomes had a unique member of this family (Lp: 11,932 pangenome families and 2,326 strict-persistent pangenome families, Ab: 31,103 pangenome families and 2,629 strict-persistent pangenome families).

## Competence genes presence/pseudogenization

We gathered from the literature a list of competence genes and genetic elements (protein coding genes, sncRNA) involved in the transformation process (S13 Data). We checked in all our strains for the presence of these elements. We retrieved their protein sequences from Paris and Philadelphia strains for *Lp* and from *A. pittii* PHEA-2 and *A. baumannii* D1279779 for *Ab*, respectively. We searched for homologous regions to proteins involved in transformation using *tblastn v.2.12.0* (-evalue 0.001, -seg no) in the genomes in our collection [76]. We deemed a gene *present* if the alignment given by tblastn had more than 80% identity with the query protein and covered more than 80% of it. sncRNA genes were searched with *blastn v.2.12.0* (-evalue 1e-10). A sncRNA gene was present if the alignment with the query gene had 100% identity and covered more than 90% of it.

When the protein coding gene was not present, we searched for pseudogenes. We considered that we identified a *pseudogene* of a competence gene when we identified a gene/pseudogene more than 80% identical to the query protein and with an alignment covering between 20% and 80% of the protein (using the output of tblastn as described above). Some genes may look like disrupted simply because they are at the border of a contig. To account for contig borders, we considered that if in addition the alignment was located at less than 50 nucleotides from the end of the contig, we could not classify it as present, missing or pseudogene and called them unascertainable. Some genes marked as pseudogenes are split on 2 different contigs. They may be interrupted or not in the actual genome. If both alignments are at the border of the contigs (less than 50 nucleotides from the end of the contig), this is consistent with the 2 hypotheses, and we marked them as unascertainable. If they are not both at the border, we marked them as pseudogenes.

Pseudogenes with large nucleotide insertions were further characterized to assess if the disruption was due to MGE and/or AMR genes. A supplementary step to assess the nature of the interruption was necessary when these pseudogenes were split on 2 contigs. We took the 2 alignments of the gene (one in each contig) and ordered them in relation to the known full gene sequence. This allowed to identify the sequence interrupting the gene. This was only the case for ComM protein, which has 3 domains organized in the following order: chlI (PF13541.9), Mgch (PF01078.24), and MgchC (PF13335.9). We relied on the coordinates and order of the domains in our alignments to identify the correct region of interruption in which we would search for MGEs and AMR genes. To specifically identify if an AbaR was interrupting *comM*, we searched for the conserved sequences any AbaR contain in its terminal region: the left-end conserved sequence ($CS_L$) and the right-end conserved sequence ($CS_R$) that corresponds to *tniC* and *tniA* presence and *orf4* presence, respectively [77]. To do so, we searched for homologous regions to TniC (WP_000736404.1), TniA (WP_000573062.1), and Orf4

(WP_001144958.1) using *tblastn* (*blast+/2.10.1*) in *comM* region of interruption. Each gene was deemed present if the alignment had more than 80% identity with the query protein and covered more than 80% of it. We deemed an AbaR being present in the genome when at least 1 conserved sequence out of $CS_L$ and $CS_R$ was identified. We did not observe any interruption of *comM* in Lp. We nevertheless searched for the Tn7-like AbaR backbone ($CS_L$ and $CS_R$) in Lp genomes with the same method but relaxed the filtering constraints (-evalue 1e-03). No Tn7-like elements were found in Lp.

In *Acinetobacter baumannii*, we had to devise a specific method to search for pilA because of the high variability of its sequence across the species [78]. We searched the most conserved region of pilA, the pilin domain (PF00114) in the representative sequence of each gene family with *hmmsearch* (*hmmer/3.3.2*). We considered that the pilin was present when the sequence score of the alignment was above the gathering threshold score cutoff (—cut_ga). Every gene family (34) presenting this domain was considered encoding PilA and every strain having this gene family was deemed having *pilA*. We checked if the strains lacking these gene families could actually have an even more diverging sequence for *pilA*. We searched for homologous regions of 4 PilA proteins (QNT88830.1, WP_000993715.1, WP_031953428.1, WP_000993729.1) previously studied in *A. baumannii* [78] using *tblastn v.2.12.0* (-evalue 0.001, -seg no) in the genomes of those strains. We relaxed the constraints compared to other genes and considered the gene present if the alignment with the query had more than 40% identity, 20% coverage, and a total length between 100 and 200 amino acids.

We also searched for the presence of the rocRp gene in the complete genomes of Lp from RefSeq downloaded on May 2023 using *blastn v.2.12.0*. We deemed it present when the same criteria as above were met (100% identity, 90% coverage). To check that chromosomes do lack rocRp, we made a complementary analysis using a lower threshold of identity (90%). This analysis also failed to reveal chromosomal versions of the gene.

## Variant calling and estimation of genetic linkage

We identified single-nucleotide polymorphisms (SNPs) and small insertions and deletion (indels) in our strains using as a reference genome the Paris strain for *Lp* and the AB5075 strain for *Ab*. The identification was done using *snippy v.4.6.0* (https://github.com/tseemann/snippy). We annotated them and predicted their functional effects with *snpEff v.4.3* [79].

From the previous variant calling, we only kept biallelic SNPs that were polymorphic in transformable strains and in non-transformable ones. We defined nonoverlapping windows of 500 bp that scanned the whole genome. We considered pairs of SNPs (A, B). In each pair, we calculated $p_r$ the frequency of the reference allele of SNP A ($A_{ref}$) and $q_r$ the frequency of the reference allele of SNP B ($B_{ref}$). We also calculated $x_{rr}$ the allelic frequencies of each pair of SNPs ($A_{ref}$, $B_{ref}$). We were then able to compute the D measure of linkage disequilibrium, its normalized value D' and the $r^2$, the square of the correlation coefficient of each pair of SNPs as follows [80].

$$D = x_{rr} - p_r q_r$$

$$D' = \begin{cases} \dfrac{x_{rr} - p_r q_r}{D_{max}} & \text{if } D > 0 \\[2ex] \dfrac{x_{rr} - p_r q_r}{D_{min}} & \text{if } D < 0 \end{cases}$$

$$r^2 = \frac{D^2}{p_r(1 - p_r)q_r(1 - q_r)}$$

with

$$D_{max} = \min(p_r(1 - q_r), (1 - p_r)q_r)$$

$$D_{min} = \max(-p_r q_r, -(1 - p_r)(1 - q_r))$$

We compared the distribution of $r^2$ in each genomic window between transformable and non-transformable populations. To assess if there was any difference in genetic linkage along the genome between transformable and non-transformable strains, we computed an average $r^2$ for each window in both populations and tested with a Wilcoxon rank-sum test if the difference between the two, $\Delta r^2$, was positive, i.e., if we rejected the null hypothesis $H_0$, $\Delta r^2 \leq 0$.

$$\Delta r^2 = r^2_{mean}(NT) - r^2_{mean}(T)$$

## Phylogenetic inference and analysis of recombination

We took the gene families that were regarded as persistent in Ab and Lp. In each species, we aligned each gene family with *mafft v.7.467* [81] within *PanACoTA v.1.2.0* (default parameters) [75] at the protein level. We back translated protein alignments to nucleotide ones (i.e., we replaced each amino acid by its original codon) because the latter provide more signal when one studies polymorphisms at the species level. Alignments at nucleotide level were concatenated to make 2 matrices of alignments of 2,629 strict-persistent genes in Ab and 2,325 strict-persistent genes in Lp ordered according to the persistent genes order and orientation of a complete genome of the collection (Lp: Paris strain; Ab: AB5075 strain). If a persistent gene was missing from the complete genome, its position and orientation was inferred from the most often frequent position and orientation it had in the collection. These matrices were then used as an input to *IQTree v.1.6.12 modelfinder* [82–84] to build the phylogenetic tree of each species.

We made a subsequent step of phylogenetic inference to account for the presence of recombination. We took the previous tree and used it as a starting point for *Gubbins v.2.4.1* [85]. *Gubbins v.2.4.1* allowed us to mask the regions in the alignment whose polymorphism was due to recombination. We finally built the phylogenetic trees based on the recombination region-free alignment with *IQTree v.1.6.12 modelfinder* according to which the best-fit model based on BIC was TVM+F+I+G4. To ensure of the branch robustness, we performed a 1,000 ultrafast bootstrap.

Trees were rooted based on outgroups: *L. longbeachae* for *L. pneumophila* and *A. baylyi ADP1* for *A. baumannii.* We added each outgroup to their respective collection. We built their pangenomes, defined their strict-persistent genomes, and aligned them in the same manner we did for the initial collections and with the same parameters. We finally built the phylogenetic tree based on this alignment with *IQTree v.1.6.12 modelfinder* (best-fit model: TVM+F+I +G4; 1,000 ultrafast bootstraps) so as to determine the position of the root in the recombination-free phylogenetic tree. All trees are listed in S1 Data.

We compared the changes in topology and branch lengths between the recombination-unaware and the recombination-free phylogenetic tree by measuring their weighted Robinson–Fould (wRF) metric (*phangorn* R package: *wRF.dist* function). This metric counts the minimum number of branch rearrangements needed to transform one tree into another and weights each branch by its length.

The average root-to-tip distance of the phylogenetic tree was calculated by averaging the distance to the root of each tip (*adephylo* R package: *distRoot* function).

Of note, Gubbins was initially intended to be used for the analysis of very closely related strains to identify recent events of recombination [85]. In this study, we used it mostly to

characterize recent events of recombination. FastGear is a software specifically developed to identify ancient events of recombination [86]. Its publication study shows that for recent events of recombination there is no significant advantage in using FastGear or Gubbins. Since FastGear is not adapted to the study of such large-scale genome data, we used Gubbins throughout our work. To assess the robustness of our key findings we compared our results when using the recombination-corrected phylogeny (the standard approach used everywhere except when explicitly stated across the main text), the tree without correction for recombination, and a third analysis where recombination was inferred with Gubbins from randomly ordered multiple alignments (in Ab). These different methods seem to have relatively little impact on the results (S17 Fig).

We also used the information on recombination tracts covering the persistent genes to assess the likelihood that neighboring non-persistent genes arose or were affected by recombination.

### Phylogenetic signal

We looked for phylogenetic inertia of the transformation trait in the rooted recombination-free phylogenetic trees. We calculated the phylogenetic signal on the log10-transformed transformation rates with Pagel's λ and Blomberg's K (*phytools* R package: *phylosig* function with test = TRUE to conduct a hypothesis test of K or λ) [87] and on the binary transformation phenotype with Fritz and Purvis' D statistic (*caper* R package: *phylo.d* function and its default parameters) [33].

### Model of evolution for the trait

We assessed the models of the quantitative evolution of the trait. For this we used the log10-transformed transformation rates across the rooted recombination-free phylogenetic trees. We tested all the models mentioned in S4 Data for the trait evolution with *fitContinuous* function from *geiger* R package [88] and with *fit_reml_levy* function from *pulsR* R package [34]. We used the $AIC_c$ as the selective criterion for the quality of model fit.

### Ancestral reconstruction of the transformation trait

We reconstructed the ancestral states of the binary transformation trait along the recombination-free rooted phylogenetic tree using the MAP prediction method and the F81 evolutionary model of *PastML v.1.9.34* [89]. We only kept nodes for which one of the possible reconstructed states had a marginal posterior probability superior to 0.6. We then assigned to the node the transformation state with the highest marginal posterior probability.

### Plasmid identification in draft assemblies

We classified contigs as plasmid by calculating the weighted gene repertoire relatedness (wGRR) of each contig against each RefSeq plasmid genome. We first searched for sequence similarity between all of their proteins using blastp (*blast+ v.2.12.0*). For each pair of contig/plasmid, the wGRR takes into account their number of bidirectional best hits and their sequence identities and gives an assessment of their gene repertoires similarity with the following formula:

$$wGRR = \sum_i \frac{id(A_i, B_i)}{min(A, B)}$$

For a pair of elements A and B, $id(A_i, B_i)$ is the sequence identity of the i-th pair of homologous

proteins. Min(A,B) is the number of proteins in the smaller element. The wGRR of 2 elements is thus defined as the sum of the identity for all pairs of homologous proteins normalized to the number of proteins found in the smallest element.

Contigs with a wGRR superior to 0.9 were kept and preliminarily assigned to the corresponding plasmid. These contigs could either be from a plasmid or be a part of the chromosome carrying a MGE similar to the plasmid. Contigs respecting the wGRR criterion, shorter than 500 kb (the maximum size of Ab and Lp RefSeq plasmids being around 300 kb) and carrying less than 50% of persistent genes were deemed part of a plasmid (S18 Fig). In Lp (Ab), 97.4% (43.3%) of the plasmids had no persistent gene at all. If the total length of the contigs assigned to the plasmid in a sample was longer than 40% of the actual plasmid length, we took this as indication of the presence of the whole plasmid in the sample. Contigs longer than 500 kb and with at least twice the number of proteins than the plasmid it was assigned to were deemed to be part of the bacterial chromosome.

We encountered in Ab 2 plasmids, Ab TG29392 plasmid pTG29392_1 from Ab TG29392 strain and plasmid pVB2107_2 from Ab VB2107 strain, which were completely integrated into chromosomal contigs (hits covering more than 50% of the plasmid). pTG29392_1 was integrated in one contig in 14% of the samples and pVB2107_2 in one contig in 3% of the samples. We classed these contigs as chromosomal contigs.

## Identification of MGEs and defense systems

We characterized the different types of mobile genetic elements. Conjugative elements were searched using *Macsyfinder 20221213.dev* [90,91] and the *CONJScan/Chromosome* model (https://github.com/macsy-models). We detected conjugative systems (CONJ) which are complete conjugative systems formed by a T4SS, a type IV coupling protein (t4cp) and a relaxase, mobilizable systems (MOB) which are systems that have a relaxase that is either alone or co-localizing with a CONJ component but not enough of the latter to make a functional conjugative system. We checked if 2 contigs of a sample presented complementary incomplete CONJ on their extremities. If so, we included them with the other conjugative systems.

Available methods cannot precisely delimit the ICEs. Hence, we defined them as regions at a distance of less than 10 kb from a conjugative system on the contigs that we did not identify as plasmid-like (cf. Plasmid identification in draft assemblies). This is a conservative estimate, given the average size of ICEs of Proteobacteria: 58 kb for $MPF_T$, 84 kb for $MPF_G$, and 87kb for $MPF_F$ [92].

Insertion sequences were searched using *ISEScan v.1.7.2.3* [93]. We used the option—remove-ShortIS to remove incomplete IS elements that is to say IS elements of length inferior to 400 bp or single copy IS element without perfect terminal inverted repeat.

Putative prophages were initially searched using *VirSorter v.2.2.3* (docker://jiarong/virsorter:latest) [94]. We further analyzed the resulting elements with *CheckV v.0.7.0* [95] and kept the ones deemed of high and medium quality. These putative viral regions were then annotated using *PHROG v.4* [96]. We built the hmm profiles of each PHROG based on their multiple alignments with *hmmbuild* (*hmmer/3.3.2* default parameters). We searched for those profiles in the previous viral regions with *hmmsearch* (*hmmer/3.3.2* default parameters). We also calculated a viral quotient for each PHROG annotation with the following formula:

$$VQ = \frac{\text{PHROG hits in phages (all RefSeq)}}{\text{PHROG hits in phages (all RefSeq)} + \text{PHROG hits in bacterial chromosome without prophages (all RefSeq)}}$$

Gene families with a viral quotient superior to 0.9 were specific to phages and were as such labeled as phage-specific genes. The final list of prophages was obtained from the preliminary

list by including information on the core genes (that should not be within prophages) and on the PHROG hits (which should be in prophages). Each putative prophage comprised between 2 persistent genes in a putative prophage containing at least 5 genes among which one had a PHROG annotation and whose average viral quotient is more than 0.5 was deemed to be a prophage.

Integrons were searched for using *IntegronFinder v.2.0.2* [97–100].

We also characterized defense systems. CRISPR-Cas systems were searched using *CRISPR-CasFinder v. 4.2.20* [90,91,101,102]. We retained the ones verifying the following criteria: the presence of a Cas gene in the corresponding genome, the presence of more than 2 spacers in the array, and imposing a mean size for spacers of at least 15 bp. We identified the targets of these spacers by running *blastn* and its task option of *blastn-short* (*blast+ v.2.12.0*) on our collection of genomes, available RefSeq plasmids and GenBank phages setting the same parameters as *CRISPRTarget* (*CRISPRSuite*) [103]. We searched for other potential defense systems with the *DefenseFinder* models v.0.0.3 [62] using *Macsyfinder 20221213.dev* [91]. This version searched for 109 defense systems.

A summary of the amount of MGEs each strain carries can be found in S14 Data.

## Presence of MGEs in recombination regions

The analysis of *Gubbins* to detect recombination was made on the persistent genes (present in most genomes, see above). To know if recombined regions potentially encompass MGEs (which are never persistent genes), we had to devise a different method. First, we counted for each persistent gene how many times it was part of a recombination tract, thus obtaining a measure of their recombination rate. In parallel, we identified the closest upstream and downstream persistent genes for every characterized MGE. Of note, we could not determine the flanking persistent genes of 39.1% (30.0%) of the MGEs because none was present in one of the directions in the contig. Among these, 9.5% (24.2%) were on contigs classified as plasmids in Ab (Lp). These elements were excluded from the analyses of genomic neighborhood. We then separated the persistent genes flanking MGEs from the others. Finally, we compared the frequency of recombination of persistent genes of these 2 groups using a Wilcoxon test. We classed the loci of MGE integration into several categories: conjugative systems, phages, ISs, and defense systems hotspots.

## Genome-wide association study

We performed a genome-wide association study (GWAS) to identify genetic determinants of transformation. We used a unitig-based approach (GWAS$_U$). A unitig is an unambiguous combination of k-mers that represents the sequences without branch in the assembly graph. This approach allows to encompass information: on SNPs and large and small indels. We can thus test all these different levels of genetic variants all at once in the GWAS$_U$. This was done on the binary transformation phenotype (TF$^{bin}$) using respectively 786 *L. pneumophila* genomes and 496 *A. baumannii* genomes with *pyseer v.1.3.9* [104]. The unitigs were called on our collection of draft assemblies with *unitig-caller* (from conda installation of *pyseer*). The association between the unitig or the gene presence/absence and the transformation phenotype was assessed with a LMM. The recombination-free phylogenetic tree allowed us to generate a distance and a kinship matrix using the scripts coming with *pyseer*, which were used in the LMM to control for population structure. We generated the QQ-plots of this analysis (S9 Fig) with *qq_plot.py* script provided by *pyseer* to verify if it was properly corrected notably for population structure. To address the problem of multiple comparisons in our analysis, we applied a Benjamini–Hochberg adjustment on the *p*-value of association already adjusted for

population structure [105]. We deemed the association between the presence of a unitig (or a gene) and the transformation phenotype significant when the adjusted *p*-value was inferior to 0.05. As a matter of comparison, we also computed the Bonferroni threshold of significancy with scripts provided by *pyseer*. This Bonferroni correction is made with the number of unique variant patterns as the number of multiple tests. The use of Bonferroni correction with the number of unique variant patterns (Lp: 406151; Ab: 357728) gave very stringent significancy thresholds (Lp: $1.23 \times 10^{-7}$, Ab: $1.4 \times 10^{-7}$). With this correction only 1 gene family association, which was negative, remained significant in Ab (and not ComM which was shown to modulate transformation rates experimentally and is correctly identified by the FDR analysis) while in Lp 65 negative and 6 positive associations stayed significant.

To study the functions of genes with unitigs that were deemed significantly associated with changes in transformation rates, we first identified the genes having the unitigs with the *annotate_hits_pyseer.py* script provided by *pyseer*. To get a clearer picture of the associations, we summarized the characteristics of each gene association over all the unitigs mapping the gene, notably its statistical significance and its positive or negative effect on the phenotype. We only consider genes that are mapped by at least 1 significant unitig. These genes can be mapped by unitigs with significant effect of different signs (positively and negatively associated with transformation), in which case we exclude them from the main analysis and study them separately (see below). We are then left with genes mapped by unitigs (often many) for which those significantly associated with transformation are all of the same sign.

Following *pyseer summarize_annotations.py*, we summarized the positive or negative association between these gene families containing the unitigs and the phenotype, its effect size (by averaging it over all the unitigs of the significant effect sign that mapped the gene), its significance (by assigning the *p*-value of the most-significantly associated unitig with the gene), its minimum allele frequency and the number of unitigs mapping them. We added on top of the functional annotation the information on the presence of the gene family in an MGE or a secretion system (as defined above). In the case of *Lp*, we also studied the unitigs in the light of the presence of sncRNA.

For the unitigs that were annotated as carrying a RM system in Ab and significantly associated with the transformation phenotype, we additionally verified if this was the system described by Vesel and colleagues [43] in A118 strain. To do so, we extracted the unitigs that mapped to the restriction modification systems. We then performed a nucleotidic alignment of these unitigs to the coding sequences composing this specific RM system, that is to say H0N27 10820 (methylase), H0N27 10825 (helix-turn helix transcriptional regulator), and H0N27 10830 (restriction endonuclease), with *blastn* (—task blastn, *blast+/2.10.1*). There was no identity between these unitigs and the previously listed components. This lack of identity suggests that we are not looking at the same RM-systems. However, we cannot exclude that the RM systems pointed out by the unitigs may have a similar mode of action of this RM system. Indeed, the mode of action of Vesel and colleagues' RM system does not rely on the methylation marks of the non-self DNA but rather consists in adding a methylation mark to the self-DNA and protect it from the endonuclease action while the non-self DNA without the mark can be degraded.

## Detection of genes with variants of opposite effect on transformation

Some gene families were associated in an ambiguous way with the transformation phenotype. In Ab (Lp) 12% (43%) of the genes with 1 unitig significantly associated with variation in transformation rates also had unitigs whose effect was significant and of opposite sign. We treated those genes separately. Indeed, such a situation could be explained by unitigs carrying

variants of the same gene (or same type of MGE) differently associated with the phenotype. So, averaging the effect size as we did before over all the unitigs of the gene would give misleading information on the association (e.g., if 2 unitigs have effects of symmetrical amplitude the average effect is zero, like for genes lacking unitigs altogether). We mapped the unitigs on the genes and called the variants with *snippy v.4.6.0* (https://github.com/tseemann/snippy). We reduced the regions of interest to the regions that overlapped between unitigs of opposite association when this was possible. We summarized the positive or negative association to the phenotype, the effect size, the significance, the minimum allele frequency, and the number of unitigs mapping to each variant. This procedure identified 52% (out of the 12%) of the genes in Ab and 66% (out of the 43%) in Lp. These genes with overlapping unitigs of opposite sense will be candidates for future analyses. Given their abundance, we further enquired on the regions with such unitigs in Lp. In Lp, half of them were in gene families of ICEs.

## Strains specificities affecting/affected by transformation rates

To characterize the association between traits across the tree, we need to take into account the phylogenetic structure of the bacterial population. We used phylogenetic logistic regression (*phyloglm* function from *phylolm* R package) [44] to estimate the effect diverse factors from our data could have on the transformation phenotype. For the phylogenetic correlations to take into account the phylogeny, we provided the recombination-free rooted phylogenetic tree and fitted our data using the "logistic_MPLE" method with the following parameters: btol = 10 and boot = 100. Confidence intervals (alpha = 0.05) were computed from the estimates of the stderr obtained given by phyloglm. All the test results can be found in S15 Data.

## Search for serine and tyrosine recombinase in RocRp-carrying plasmids

We searched for tyrosine (PF00589) and serine recombinases (presence of both PF00239 and PF07508) with *hmmsearch* (*hmmer/3.3.2* —cut_ga). No tyrosine recombinase nor serine recombinase were found.

## Search for Xer-mediated recombination in RocRp-carrying plasmids

We searched for homologous regions to the 29-nucleotide conserved sequence of the Lp *dif* site [106] in rocRp-carrying plasmids using *blastn* (—task blastn-short, *blast+/2.10.1*). We found no hits with more than 80% identity and more than 80% coverage of the queried sequence). We also searched for XerL orthologs in rocRp-carrying plasmids. We searched for the XerL sequence from Lp strain Lp02 [106] in the plasmid genomes using *tblastn* (*blast +/2.10.1* -evalue 1e-03). We found no hits to this sequence in the RocRp-carrying plasmids.

## Structural prediction of hypothetical proteins of interest

To get insights into the functional role of proteins of unknown function, we characterized their protein domains. We first predicted their protein structure with *ColabFold v.1.5*. [107]. The output protein structure was then given to *Foldseek* server [108] which compared it to a large collection of protein structures (6 databases: AlphaFold/Proteome, AlphaFold/Swiss-Prot, AlphaFold/UniProt50, GMGCL, MGnify-ESM30, PDB100) in 3Di/AA mode.

## Graphical representation of GWAS output

All the graphs were done with *R v.4.1.0*.

## Supporting information

**S1 Data. List of phylogenetic trees computed in this study and their features.**
(XLSX)

**S2 Data. S2A Data.** Measures of the transformation rates of the transformable and non-transformable control strains in *Legionella pneumophila*. **S2B Data.** Measures of the transformation rates of the transformable and non-transformable control strains in *Acinetobacter baumannii*.
(XLSX)

**S3 Data. Measures of phylogenetic signal on continuous transformation rates.**
(XLSX)

**S4 Data. Evolutionary models tested on the log10-transformed transformation rates and their fitness in *Legionella pneumophila* and *Acinetobacter baumannii*.**
(XLSX)

**S5 Data. Association of transformation phenotype and change with recombination features tested with Wilcoxon tests.**
(XLSX)

**S6 Data. S6A Data.** GWAS results obtained with a unitig approach on the binary transformation phenotype in *Legionella pneumophila*. **S6B Data.** GWAS results obtained with a unitig approach on the binary transformation phenotype in *Acinetobacter baumannii*. **S6C Data.** GWAS results obtained with a unitig approach on the binary transformation phenotype in *Legionella pneumophila* with rocRp as covariate.
(XLSX)

**S7 Data. Reproducibility of the transformation luminescence assay in *Legionella pneumophila* and *Acinetobacter baumannii*.**
(XLSX)

**S8 Data. Collection of *Legionella pneumophila* and *Acinetobacter baumannii* strains and their measures of transformation rates.**
(XLSX)

**S9 Data. Cumulated lengths of the recombination tracts (log10-transformed) depending on phenotype change in *Legionella pneumophila* and *Acinetobacter baumannii*.**
(XLSX)

**S10 Data. Difference between transformable and non-transformable strains of their squared correlation (r2) between bi-allelic values at 2 loci in windows of 500 nt along their genomes.**
(XLSX)

**S11 Data. Quantification of the variations in expression of the nanoluc marker in randomly chosen strains.**
(XLSX)

**S12 Data. S12A Data.** Presence of genes involved in natural transformation in *Legionella pneumophila*. **S12B Data** Presence of genes involved in natural transformation in *Acinetobacter baumannii*.
(XLSX)

**S13 Data. S13A Data.** Protein-encoding genes involved in natural transformation in *Legionella pneumophila*. **S13B Data.** sncRNA genes involved in natural transformation in *Legionella*

*pneumophila*. **S13C Data.** Protein-encoding genes involved in natural transformation in *Acinetobacter baumannii*.
(XLSX)

**S14 Data. S14A Data.** MGE burden per strain in *Legionella pneumophila*. **S14B Data.** MGE burden per strain in *Acinetobacter baumannii*.
(XLSX)

**S15 Data. Statistical tests performed throughout the study comparing transformable and non-transformable strains in *Legionella pneumophila* and *Acinetobacter baumannii*.**
(XLSX)

**S16 Data. S16A Data**. Absolute variation of transformation rates between 2 strains depending on their patristic distances in *Legionella pneumophila*. **S16B Data**. Absolute variation of transformation rates between 2 strains depending on their patristic distances in *Acinetobacter baumanni*.
(XLSX)

**S17 Data. S17A Data.** Regressions of the variations of strain transformation rate depending on its patristic distance with the donor strain in *Legionella pneumophila*. **S17B Data.** Regression of the variations of strain transformation rate depending on its patristic distance with the donor strain in *Acinetobacter baumannii*.
(XLSX)

**S18 Data. Variations of strain transformation rate depending on its patristic distance with the donor strain in *Acinetobacter baumannii* and *Legionella pneumophila*.**
(XLSX)

**S19 Data. Log10-transformed transformation rates depending on the percentage of identity shared between the homology arms of the strain and the plasmid donor strain ones in *Acinetobacter baumannii* and *Legionella pneumophila*.**
(XLSX)

**S20 Data. Distribution of pseudogenized comM and the nature of its interruption if interrupted in *Acinetobacter baumannii*.**
(XLSX)

**S21 Data. S21A Data.** Data set used to generate QQplot in *Legionella pneumophila*. **S21B Data.** Data set used to generate QQplot in *Acinetobacter baumannii*.
(XLSX)

**S22 Data. Relatedness of *Legionella pneumophila* Refseq.**
(XLSX)

**S23 Data. Relationship between the number of transformants and the luminescence measured in *A. nosocomialis* M2 strain.**
(XLSX)

**S24 Data. Benchmarking the luminescence assay by comparing the transformation frequencies determined by selection with a genetic marker and by luminescence values (RLU) in *A. nosocomialis* M2 strain.**
(XLSX)

**S25 Data. Variations of transformation rates measured by the luminescence assay in well-documented strains.**
(XLSX)

**S26 Data. Terminal branch lengths depending on the type of alignment (reference/random).**
(XLSX)

**S27 Data. Recombination rates of genomes depending on the type of alignment (reference/random).**
(XLSX)

**S28 Data. Cumulated lengths of recombination tracts in alignment concatenated according to the reference or randomly.**
(XLSX)

**S29 Data. Proportion of persistent genes in plasmids.**
(XLSX)

**S1 Fig. Evolutionary theories behind natural transformation's persistence.**
(DOCX)

**S2 Fig.** Distribution of the log10-transformed transformation rates in *Acinetobacter baumannii* (left) and *Legionella pneumophila* (right) non-transformable (NT) and transformable (T) control strains.
(DOCX)

**S3 Fig.** Absolute variation of transformation rates between 2 strains depending on their patristic distances in *Acinetobacter baumannii* (left) and *Legionella pneumophila* (right).
(DOCX)

**S4 Fig.** Variation of strain transformation rate depending on its patristic distance with the donor strain in *Acinetobacter baumannii* (left) and *Legionella pneumophila* (right).
(DOCX)

**S5 Fig.** Log10-transformed transformation rates depending on the percentage of identity shared between the homology arms of the strain and the plasmid donor strain ones in *Acinetobacter baumannii* (left) and *Legionella pneumophila* (right).
(DOCX)

**S6 Fig. Schematic representation of the models of diverse evolutionary dynamics fitted to the transformation rates (adapted from Landis and colleagues [34]).**
(DOCX)

**S7 Fig. Difference between transformable and non-transformable strains of their squared correlation ($r^2$) between bi-allelic values at 2 loci in windows of 500 nt along their genomes.**
(DOCX)

**S8 Fig. Distribution of pseudogenized *comM* and the nature of its interruption if interrupted across the phylogenetic tree of *Acinetobacter baumannii* depending on the source of the strain.**
(DOCX)

**S9 Fig.** QQ-plots generated from the outputs of the unitig-based Genome-Wide Association Study on the binary transformation phenotype in *Acinetobacter baumannii* (left) and

*Legionella pneumophila* (right).
(DOCX)

**S10 Fig.** Alignment of the structure of the DarA_N domain-containing protein in *A. baumannii* ACICU (A0A4Y3J949 UniProt) (yellow) with the structure of the most significantly associated protein (pangenome family 6781) with non-transformability carried by prophages (blue) in *Acinetobacter baumannii*.
(DOCX)

**S11 Fig. Relatedness of *Legionella pneumophila* Refseq plasmids and presence of the sncRNA rocRp in their genomes.**
(DOCX)

**S12 Fig. Volcano plots showing average effect sizes and significance of the association of the gene families with the transformation phenotype according to $GWAS_U^{bin}$-cov in Legionella pneumophila and plasmid association with the phenotype**
(DOCX)

**S13 Fig. Linear relationship between the number of transformants and the luminescence measured in *A. nosocomialis* M2 strain.**
(DOCX)

**S14 Fig. Benchmarking the luminescence assay by comparing the transformation frequencies determined by selection with a genetic marker and by luminescence values (RLU) in *A. nosocomialis* M2 strain.**
(DOCX)

**S15 Fig. Variations of transformation rates measured by the luminescence assay in well-documented strains.**
(DOCX)

**S16 Fig. Reproducibility of the luminescence assay across replicates in *Acinetobacter baumannii* and *Legionella pneumophila*.**
(DOCX)

**S17 Fig. Comparison of the different phylogenies inferred in *Acinetobacter baumannii* based on different evolutionary features.**
(DOCX)

**S18 Fig.** Distribution of the proportion of persistent genes in contigs assigned as plasmids in *Acinetobacter baumannii* (left) and *Legionella pneumophila* (right).
(DOCX)

## Acknowledgments

We thank Eloïse Guignard for technical assistance in isolation of *L. pneumophila* isolates. We thank Evelyn Skiebe for preparing and shipping *A. baumannii* isolates and members of the RKI central sequencing lab for excellent technical assistance. We thank Eugen Pfeifer for his help in defining and providing all the data necessary for the viral quotient calculation. We are grateful to Suzana Salcedo for providing additional *A. baumannii* isolates [109]. The strains CIP and CRBIP have been obtained from the Collection Institut Pasteur (CRBIP, Paris, France). This work used the computational and storage services (TARS cluster) provided by the IT department at Institut Pasteur, Paris.

## Author Contributions

**Conceptualization:** Fanny Mazzamurro, Xavier Charpentier, Eduardo P. C. Rocha.

**Data curation:** Fanny Mazzamurro, Jason Baby Chirakadavil, Isabelle Durieux, Julie Plantade, Christophe Ginevra, Sophie Jarraud, Gottfried Wilharm.

**Formal analysis:** Fanny Mazzamurro, Julie Plantade.

**Funding acquisition:** Xavier Charpentier, Eduardo P. C. Rocha.

**Investigation:** Fanny Mazzamurro, Jason Baby Chirakadavil, Ludovic Poiré, Xavier Charpentier, Eduardo P. C. Rocha.

**Methodology:** Fanny Mazzamurro, Jason Baby Chirakadavil, Xavier Charpentier, Eduardo P. C. Rocha.

**Project administration:** Xavier Charpentier, Eduardo P. C. Rocha.

**Resources:** Jason Baby Chirakadavil, Isabelle Durieux, Christophe Ginevra, Sophie Jarraud, Gottfried Wilharm, Xavier Charpentier.

**Supervision:** Xavier Charpentier, Eduardo P. C. Rocha.

**Validation:** Fanny Mazzamurro, Xavier Charpentier, Eduardo P. C. Rocha.

**Visualization:** Fanny Mazzamurro, Eduardo P. C. Rocha.

**Writing – original draft:** Fanny Mazzamurro, Jason Baby Chirakadavil, Eduardo P. C. Rocha.

**Writing – review & editing:** Fanny Mazzamurro, Christophe Ginevra, Gottfried Wilharm, Xavier Charpentier, Eduardo P. C. Rocha.

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
