## [Editor Report · Decision Letter 0]

17 Jan 2024

Dear Dr Mazzamurro, 

Thank you for submitting your manuscript entitled "Intragenomic conflicts with plasmids and chromosomal mobile genetic elements drive the evolution of natural transformation within species" for consideration as a Research Article by PLOS Biology.

Your manuscript has now been evaluated by the PLOS Biology editorial staff, as well as by an academic editor with relevant expertise, and I'm writing to let you know that we would like to send your submission out for external peer review.

Once your full submission is complete, your paper will undergo a series of checks in preparation for peer review. After your manuscript has passed the checks it will be sent out for review. To provide the metadata for your submission, please Login to Editorial Manager (https://www.editorialmanager.com/pbiology) within two working days, i.e. by Jan 19 2024 11:59PM.

Kind regards,

Roli Roberts

Roland Roberts, PhD

Senior Editor

PLOS Biology

rroberts@plos.org

---

## [Decision Letter · Decision Letter 1]

4 Apr 2024

Dear Dr Mazzamurro,

Thank you for your patience while your manuscript "Intragenomic conflicts with plasmids and chromosomal mobile genetic elements drive the evolution of natural transformation within species" was peer-reviewed at PLOS Biology. It has now been evaluated by the PLOS Biology editors, an Academic Editor with relevant expertise, and by four independent reviewers. 

You'll see that reviewer #1 calls the study “excellent,” but has four major requests; he wants you to use a different recombination rate algorithm, to quantify the effects of restriction enzymes, to consider explanations for the presence of RocRp RNA on plasmids, and to consider an alternative explanation for high recombination around MGEs. Reviewer #2 is also positive, but, like rev #1, wants you to use fastGEAR instead of Gubbins (though for a different reason), to use a more stringent p-value for the GWAS (and do a QQ-plot), and to use a different threshold to distinguish transformable and non-transformable strains. Reviewer #3 says s/he enjoyed reading the paper, but wants you to make the text more concise, to add a model figure, to validate your luminescence assay, and questions how you assessed strain divergence when correlating with luminescence. Reviewer #4 is concerned about the possible effect of growth conditions on the large-scale transformation assay, suggesting that you test this on a small number of isolates; like rev #3, s/he finds the writing challenging for the PLOS Bio readership, and also suggests validating some of the GWAS hits by KO or transgenesis.

In light of the reviews, which you will find at the end of this email, we would like to invite you to revise the work to thoroughly address the reviewers' reports.

Given the extent of revision needed, we cannot make a decision about publication until we have seen the revised manuscript and your response to the reviewers' comments. Your revised manuscript is likely to be sent for further evaluation by all or a subset of the reviewers.

**IMPORTANT - SUBMITTING YOUR REVISION**

*Re-submission Checklist*

*Published Peer Review*

*PLOS Data Policy*

*Blot and Gel Data Policy*

Sincerely,

Roli Roberts

Roland Roberts, PhD

Senior Editor

PLOS Biology

rroberts@plos.org

REVIEWERS' COMMENTS:

Reviewer #1:

This manuscript by Mazzamurro et al is an excellent study of the evolutionary benefit of transformation, which combines modelling of genomic data with high-throughput experimental work. It builds on a related previous analysis of Legionella pneumophila by some of the same authors (https://www.ncbi.nlm.nih.gov/pmc/articles/PMC6744872/). I have only a few comments that I believe should be addressed in a revised version:

Comment #1: The current analysis of recombination rates using genomic data violates the assumptions of the algorithms being used. The Gubbins analysis is run on a species-wide tree using persistent genes "concatenated in a random order". However, the creators of Gubbins describe such an input as being "fundamentally incompatible" with the software (https://sanger-pathogens.github.io/Roary/). This is because Gubbins relies on the spatial arrangement of polymorphisms in the genome, which is not accurately represented by a permutation of the persistent gene alignments. Instead, the authors should analyze their data with an algorithm such as fastGEAR (https://pubmed.ncbi.nlm.nih.gov/28199698/).

Comment #2: How much variation in transformation rate is explained by the different mobile genetic elements, and how much by defence systems such as restriction enzymes? Restriction enzymes can have a strong inhibitory effect on the acquisition of insertions by transformation (e.g. https://www.ncbi.nlm.nih.gov/pmc/articles/PMC3573125/), and therefore might strongly affect the results of the assay of transformation rates. The authors should quantify how susceptible their assay is to the effects of restriction by comparing the rates of acquisition of nanoluc in a mutant lacking a restriction system strongly associated with transformation rates, relative to rates in the parental strain.

Comment #3: The discussion of why the RocRp RNA is found on plasmids is interesting. Three possible explanations that might be considered are:

(a) These plasmids might be episomal in Lp or another host species, such that the RNA would be beneficial when they were integrated - do they carry an integrase? In the absence of an integrase, do they have a dif-like sequence (https://www.biorxiv.org/content/10.1101/2021.09.12.459815v1.full) that can exploit host recombinase activity to insert into the genome?

(b) The plasmids are highly variable in gene content (e.g. https://www.nature.com/articles/s41598-022-09721-9) - the RocRp genes may be protecting themselves from being deleted from the plasmids by homologous recombination

(c) Lp is highly unusual in not being commonly infected by prophage, but still containing many ICEs. Is it possible that plasmids benefit from mobilisation by ICEs or episomal elements that encode conjugative systems that could aid the plasmids' spread? Therefore the plasmids may shut down transformation to preserve the ICEs.

Comment #4: The authors identified elevated levels of recombination in the persistent genes flanking MGEs. However, numerous MGEs drive recombination specifically in the flanking DNA, without any role for transformation. In particular, conjugation can drive transfer of nearby DNA (e.g. https://doi.org/10.1073/pnas.0803654105), and prophage can transfer neighboring sites through specialised or lateral transduction (https://doi.org/10.1371%2Fjournal.ppat.1007878). How do the authors rule out such alternative explanations?

Minor comments:

(a) It would be helpful to include the hypothesis that transformation primarily facilitates adaptive evolution more explicitly in the Introduction, as it is mentioned in the Results

(b) Fig. 4: it would be helpful to label the polymorphisms associated with the biggest impacts in the volcano-type plots

(c) It has recently been suggested that the capsule of A. baumanii affects its transformation rate (https://www.biorxiv.org/content/10.1101/2024.02.15.580542v1). Did the authors identify any capsule loci in their analysis?

Reviewer #2:

In their manuscript, Mazzamurro et al. assembled a very interesting dataset to investigate natural transformation and its distribution within bacterial species. They did so at a relatively large scale, with hundreds of isolates being measured for two important bacterial species with different lifestyles: Legionella pneumophila (Lp) and Acinetobacter baumannii (Ab). This very useful dataset is then united with whole genome sequencing of all the isolates to ask questions about the distribution of natural transformation with respect to the phylogeny of the species, the evolution of this trait, the genetic variants associated with variation in this phenotype, and the role of mobile genetic elements. The manuscript is well written and easy to follow, and the underlying data represents, to the best of my knowledge, a unique view on the distribution of natural transformation within bacterial species in terms of the number of isolates being assessed. The "competition" between transformation and mobile elements (previously proposed on this very journal) is compelling, although difficult to conclusively prove through what is essentially an observational study confirming a previous hypothesis. The authors are however careful to note the limitations of their analyses, leaving careful readers with plenty of information to independently asses the work presented in this study.

I want to focus my critique of this study on two methodological aspects, for which I am more qualified to speak. They would necessarily need to be addressed if this manuscript is to be published in this high visibility journal.

- Use of gubbins to detect recombination events within a species: there is an ongoing discussion in the field of microbial genomics as to whether it is appropriate to apply recombination methods based on the "Clonal Frame" model across "strains". By "strain" here I indicate what is also sometimes known as a "lineage", which indicates a set of very closely related set of isolates. This limitation is acknowledged in a publication describing a competing method to detect recombination between (and within) lineages (10.1093/molbev/msx066). Regardless of where the authors stand on this debate I think they should acknowledge that the way they identify recombination may be inappropriate, and thus invalidating for many of their analyses that depend on that. But even better, the authors may try to use fastGEAR to obtain a different list of recombination events to be used to test the robustness of their subsequent analyses.

- The GWAS analysis is using a very lenient corrected p-value threshold, which leads the authors to associate a very high number of gene families to the transformation phenotype. Since they used pyseer on variants encoded by unitigs, they could follow the recommendation to use the number of unique variants presence/absence patterns to derive a more sensible p-value threshold (see: https://pyseer.readthedocs.io/en/master/usage.html#number-of-unique-patterns). I suspect that the number of significant hits will be significantly reduced, which may bring more clarity to this section. I would then also suggest that the authors generate a QQ-plot to verify that their association analysis is well corrected.

Medium point:

- L595: "The threshold between transformable and non-transformable strains was set based on the maximum of the transformation rates of a non-transformable strain". Why using an extreme value as a threshold when the authors have so many replicates for the non-transformable strains? Wouldn't it make more sense to use the mean/median/upper quartile? What if the maximum value happens to come for a replicate that gave a very high reading for technical reasons? Any change in their threshold would of course potentially affect all analyses using the derived binary phenotype, and so more clarity is needed here. Additionally, the GWAS could potentially also be carried out on the continuous phenotype, which would not be affected by this problem, although I recognize that a binary phenotype is more intuitive in this study

Minor points:

- L211 "its effect is small and non-significant when phylogeny is not accounted for" <- not sure what this means exactly

- L308: "Most of the strongest genetic determinants of transformation inhibition are in MGEs": this is not quantified

- Figure 4A: the y-axis label says "log10" but I assume the authors did a natural log transform, as ln(0.05) will yield a value of ~3. This should be corrected

- Figure 4B: it seems like most genes in Lp belong to the ICE category, which might indicate that the way they are annotated is too lenient for Lp?

- L432: what are the alternative hypotheses that are less simple than the one the authors propose?

- L465: I was a bit confused by the last sentence in the paragraph. Could the authors spell this idea out a bit better?

Reviewer #3:

Review of PBIOLOGY-D-24-00017R1

Summary: The authors present evidence that the bacterial natural competence pathway and mobile genetic elements (MGEs) are in intragenomic conflict for two bacterial species, Acinetobacter baumannii (Ab) and Legionella pneumophila (Lp). For each species, they examined hundreds of strains that they had available in the lab and for which they had draft genome sequences available, and they assayed natural transformation rates across the entire set of strains using a novel luminescence approach. They applied a series of genomic evolutionary analyses to examine patterns of how transformability varies across strains with respect to: the phylogenetic tree, population inferences of recombination among the genomes, and identification of specific genetic variants associated with differences in transformability. They found strong signatures of MGEs as associated with loss of transformability, and they argue that this fits a "chromosome curing" model for the maintenance of natural competence in bacteria over evolutionary time.

Assessment: I enjoyed reading this paper. It presents an enormous effort from both an experimental and computational perspective, and the insights the authors reach with this pair of datasets are compelling. In particular, the transformability GWAS is quite a feat and the MGE-related results therein are convincing and impactful. The writing feels a bit convoluted and wordy in places, which could make accessibility to a broader audience more challenging, so conciseness could be improved and redundancies removed to reduce the total length. A detailed model figure could help. Their model makes sense but there are places where cause-and-effect are not established, and so chicken-and-egg situations set themselves up. For example, even if there is a correlation between acquiring MGEs and loss of competence, that doesn't necessarily mean that loss of competence is why those lineages appear prone to extinction. What if it's MGE load causing a fitness problem? Similarly, MGEs do not have to directly inactivate or suppress natural competence for the model to work, since they might also be more prone to accumulate in lineages that lost competence for other reasons. Thus there is pervasive language that makes assumptions about cause and effect that are not needed or can be relegated to the discussion. I also ask the the authors to expand and detail more about their novel luminescence assay for transformation, which allowed them to assay a very large number of strains. While I think that the assay seems reasonable, there is insufficient detail to address potential artifacts that could cause their luminescence measurements to not accurately measure transformability. There are also various places where the comparison between the two species breaks down that require further explanation, along with some other more minor issues.

Comments:

1. The authors need to show validation of their luminescence assay. Only showing a wild type and a transformation null mutant is okay but corroborative transformants/CFU measures as in traditional assays for at least 3-4 genetically distinct isolates with varying intermediate transformation rates, ideally more, is needed (also, can comM be added?). Surely the authors already have these data. There are a variety of potential artifacts that might arise in the context of assaying a diverse set of strains within a species. The authors controlled for differences in biomass by normalizing luminescence to the optical density of the cultures at the time of the assay, but this does not discount expression differences of the Nluc construct related to growth phase, natural variability in the regulation of competence, and several other possible confounders. These issues need not be perfectly addressed but a clearer demonstration that the luminescence quantitatively correlates with transformation frequency using traditional assays should be shown. Similarly, they state that the measurements are reproducible and did 3-6+ replicates, but no actual spread of LU/RLU measurements across replicates were shown except buried in Table S6, which shows pretty big differences between assay; for some strains, they have up to 120 measurements that were presumably done as controls across assays. Some additional exploration or supplementary information about potential batch effects should be addressed, since otherwise the authors only show this data in Figure 1A. Some of the overall patterns seen (e.g. jump process, etc) could also arise with data that were semi-randomized, so ensuring that these rapid transitions aren't because of non-transformation-related luminescence variability is important.

2. The authors obliquely address the issue of whether sequence divergence between the recipient chromosome and the homology arms of their transforming plasmid might drive patterns of luminescence by testing whether related strains had correlated luminescence levels. They report no relationship, but: (A) this isn't directly testing divergence between the relevant loci, and (B) the stats appear to conflict for Ab and Lp, suggesting that these organisms are not comparable in this respect, though they are treated as if they are. 

A. Divergence between a pair of strains can vary dramatically in sliding windows across a genome, especially given recombination among lineages. Thus, the correct test would be to look at the divergence between the actual homology arms (targeting a minor pilin operon for some reason) of the wild-type lab strains and the recipient strains' orthologous sequences (assuming synteny of these 2 kb arms across strains, but if not, that would also reduce transformation rates). Even identity between donor and recipient still does not show a strong correlation, this still does not exclude variable divergence of donor and recipient as making contributions to the the observed luminescence.

B. Stats on at Lines 144-146 and Figure 1A contradict the statement that variance in luminescence does not correlate with phylogenetic distance. This appears true for Ab but not Lp, and the Figure clearly shows that clades of closely related Lps have correlated luminescence values.

3. This is not a major criticism, but regarding the transformation assay, I was surprised to find that the authors are using an intact circular plasmid as donor for transformation, rather than releasing the cassette with its homology arms, which should facilitate translocation and recombination. I would expect higher rates of transformation, but perhaps there are details of these assays or these organisms that differ from some other naturally competent organisms.

4. The Lp and Ab trees are quite distinct, and I understand how the authors are using that here, however, it is noteworthy that all Lp isolates are clinical (maybe all from the same hospital system?), whereas Ab isolates were a more diverse collection of both environmental and clinical. Could this explain some of the discrepancies seen here, or is this sampling bias a potential confounder in comparing these collections? 

Other comments:

Lines 164-178: This is fine but needs a better way to motivate or explain.

Lines 180-200: Lack of caution about correlation and causation. Later, the demonstration that these lineages have a high load of MGEs gives an alternative explanation.

Lines 208-210: Comforting, but is this not slightly confounded by the bias of non-transformables to be at the tips?

Lines 213-217/Figure 2A: This presentation is strange. The "CLRT" here are logs higher than a genome size, so some form of scaling might help these numbers make more sense. It would also be useful to report n within each observation, since it's unclear how many are in each set. But most importantly, the text is noting the discrepancy between Ab and Lp but otherwise ignoring the fact that Lp did not support the stated observation.

Lines 224-226/Figure 2B: This is a nice demonstration that the non-transformable strains have experienced less recent recombination, but this does not require any allusion to whether this recombination is "adaptive".

Lines 237-271: Are missing minor pilins known to be required or only sometimes? This section again pretty much ignores Lp. What about gspHH, fimT in Lp? Might be worth acknowledging other types of variation could alter transformability, since several of these genes return in the next section.

Line 293: Can comM be pointed out on Figure 4A?

Line 308: This is a misstatement. MGEs are the most abundant but not "the strongest determinants of transformation inhibition", since there are often other stronger associations. Also, despite recombination among these strains, it is not at all clear that all these MGEs are "determining" the lower transformation frequencies; they are associated. They could be linked by being trapped in a poorly transformable genetic background, rather than directly inhibiting transformation. Some may, but this accumulation of MGEs in these backgrounds does not require that this be so. It could also be that the lack of "chromosome curing" means they're accumulating in these backgrounds without any active inhibition, as sort-of alluded to in the section starting at Line 364.

Line 409: "We found very little association of transformation rates and population structure…" in Ab, but the stats and figure for Lp suggest that at least some clades and groups of closely related strains share similar transformation rates.

Line 413-430: It is satisfying to see the slight increase in LD breakdown in transformable over non-transformable strains. This is a great observation, even though it is a weak effect. However, this discussion assumes that transformation rates would correlate with recombination rates but transformation often not lead to genetic change, so breakdowns in the correlation could be related to the availability of genetically distinct donors. E.g. competence could be high in situations where most available DNA is from a highly similar cell, and so transformation rates would be high. Another explanation for the only weak correlation is that the benefits of natural competence are only weakly connected with its effect on recombination.

Line 437: Ref 39 is to a mutant rather than a natural isolate; this might be the wrong citation. The observation with comM is really interesting but this strong affect was only in Ab. I don't think there was any evidence for this working in Lp. Why not? Is the transformation phenotype different for this mutant? The discussion in general seems to somewhat overlook Lp. What is the nature of the comM inactivations in Ab? Are they different MGEs? Are they targeting comM or accumulating there? This and the overall model might require a detailed figure.

Lines 449-453: This represents a completely distinct hypothesis for the evolution and maintenance of the competence machinery that is not listed in Figure S1 or discussed elsewhere but really does represent an important potential mechanism for the maintenance of natural competence: i.e. the machinery may have pleiotropic roles in similar or related processes, or even that interacting with extracellular DNA could have benefits beyond food or sex.

Lines 455-467: "Justifies" is the wrong word here. It is worth noting that observations in the Pasteurellaceae (in Ref 41) led Rosie Redfield to entertain the idea of "clade level" selection for competence from her much more limited data. The observations here with MGEs accumulated into non-transformable strains and that these are groups are biased to the tips of the tree. 

Lines 469-483: I don't know that the "chromosome curing" model requires that MGEs block competence in all cases but they should still be able to accumulate in non-transformable backgrounds. This is the chicken-egg thing again. Clarifying the model here is important. 

Lines 518-529/Table S6: More information about these isolates is warranted. Are any of the Lp results here affected by the sampling of only clinical isolates? Is there additional metadata to associate with this table, e.g. time frames isolates were collected, etc.

Lines 530-536: More details about nLuc and the targeting are warranted. How is nanoLuc expressed and under what control? Plasmids were used straight without cutting?

Lines 538-551: Are the quality filtered assemblies indicated in Table S6 or just excluded?

Lines 553-586: Were there corroborative measures of transformability using other assays? RLU values reported in Table S6 are highly variable. Was anything done to normalize batches to controls or other?

Lines 588-599: It is reassuring that the luminescence was normalized to OD, but were any checks done to ensure no correlation between OD and RLUs? If highly variable growth rates mean that assays were done with cells in different growth phases, this could affect interpretations substantially.

Lines 600-646: This is a good treatment but does remind me that in the GWAS results and discussion, it may be worth bringing up that many of the hits outside MGEs etc could be things that affect the regulation of competence that might be somewhat undetermined.

Lines 648: Thanks. This is useful. How does this work with the "random gene order" versus "ancestral gene order" versions of these analyses, and how is the inferred ancestral order determined?

Lines 715-737: Presumably all the interesting patterns breakdown when data are permuted or partially randomized?

Lines 739-824: Nice detailed treatment. Given all this extra annotation, it would be extremely useful if a supplement were included that provided MGE "burden" for each strain.

Lines 826-877: Great!

Reviewer #4:

In this manuscript, the authors describe variation in competence in two model species - Acinetobacter baumannii and Legionella pneumophila - and carry out extensive bioinformatics analysis to identify genetics determinants of this observed variation. This is a comprehensive study that has the potential to make an important conceptual advance to our understanding of the driving forces behind the evolution of natural competence. As a non-bioinformatics specialist, I will focus my review on the conceptual advances rather than the methodological underpinnings.

Major:

1- The large scale transformation assays in Fig. 1 are well-designed and produce exciting data that feed into the majority of the downstream analysis. Developing and optimising this high-throughput assay is extremely valuable, and reveals high levels of between strain variation in recombination when supplying DNA encoding luciferase. My main concern with this assay is that it is unclear how sensitive the results are to variation in growth conditions. For example, different strains may have evolved to switch competence on or off under different environmental conditions. If so, this could result in (many) false-negative results in the assay, with (potentially major) implications for the downstream modelling of competence gain/loss and GWAS analysis. In addition, recombination success/failure may depend on micro-homology between the luciferase DNA and the bacterial chromosome, which may not follow the phylogeny if homology is within mobile elements, or if lack of recombination is due to sequence-specific nucleases in the bacteria that move by HGT. These factors have not been sufficiently taken into account. Perhaps, the authors could carry out assays on a selected (small) number of recombination +ve and -ve isolates to examine the robustness of the Fig. 1 data to variation in growth conditions (e.g. varying nutrients, temperature, growth phase) and DNA sequence (e.g. with/without homology arms)? 

2- Transformation rates evolve as a jump process - I felt this section was rather dense and quite inaccessible to general readership of PLoS Biol. I suggest the authors to rework this part of the paper , explaining a more detailed description of the question , approach, explanation of how these models work, confidence in the models and ability to discriminate between them. Perhaps including control genes / traits for which patterns of evolution are well established would be helpful to illustrate differences in dynamics (e.g. housekeeping genes?).

3- The GWAS data point towards MGE gene candidates that inhibit competence / transformation, as well as genes that are associated with increased competence. The obvious next step will be to validate at least some of the most promising candidates by deleting them from non-competent strains and/or introducing them into competent strains, and demonstrate their effect on recombination using the assay in Fig. 1. Including such experimental validations would really make the study shine, and alleviate any concerns to do with experimental biases and downstream modelling referred to in comment 1.

Minor:

L132 The authors state that "transformation assays were performed multiple times and revealed good concordance" - it would be good to provide a more quantitative description of the variation observed, and to make these data available in SI. 

L199 Please indicate effect size for both species

L222 Delta-R^2 statistic - Please provide calculation for each species independently

L249-252 Is this analysis done using a phylogenetically-aware model? 

L308 - Please provide statistics to support claim that majority of inhibitors are MGE encoded

---

## [Decision Letter · Decision Letter 2]

15 Jul 2024

Dear Dr Mazzamurro,

Thank you for your patience while we considered your revised manuscript "Intragenomic conflicts with plasmids and chromosomal mobile genetic elements drive the evolution of natural transformation within species" for consideration as a Research Article at PLOS Biology. Your revised study has now been evaluated by the PLOS Biology editors, the Academic Editor and the original reviewers. 

You'll see that reviewer #1 has two remaining points, which are broadly, a) to do a multivariate analysis exploring the relationship between mobile elements and R-M loci, and b) some clarification around the role of RocRp. Reviewer #2 thinks that you need a more stringent p-value for your GWAS, after Bonferroni correction for multiple testing. Reviewer #3 is mostly happy and just picks you up on two minor points. Reviewer #4 is completely satisfied.

IMPORTANT: In addition to attending to the reviewers' remaining points, please address the following:

a) Please comply with PLOS' Data Policy (https://journals.plos.org/plosbiology/s/data-availability); specifically, we need you to supply the numerical values underlying Figs 1AB, 2AB, 4AB, 5, S2, S3, S4, S5, S7AB, S8, S9, S11, S12AB, S13, S14, S15, S16, S17BCD, S18, either as a supplementary data file or as a permanent DOI’d deposition.

b) Please cite the location of the data clearly in all relevant main and supplementary Figure legends, e.g. “The data underlying this Figure can be found in S1 Data” or “The data underlying this Figure can be found in https://zenodo.org/records/XXXXXXXX

c) Please make any custom code available, either as a supplementary file or as part of a data deposition.

In light of the reviews, which you will find at the end of this email, we are pleased to offer you the opportunity to address the remaining points from the reviewers in a revision that we anticipate should not take you very long. We will then assess your revised manuscript and your response to the reviewers' comments with our Academic Editor aiming to avoid further rounds of peer-review, although might need to consult with the reviewers, depending on the nature of the revisions.

**IMPORTANT - SUBMITTING YOUR REVISION**

*Resubmission Checklist*

*Published Peer Review*

*PLOS Data Policy*

*Blot and Gel Data Policy*

Sincerely,

Roland

Roland Roberts, PhD

Senior Editor

PLOS Biology

rroberts@plos.org

REVIEWERS' COMMENTS:

Reviewer #1:

I thank the authors for the time they have taken in addressing to my comments; their response is excellent. I have two remaining major comments, and some minor comments.

Major comments :-

[1] I thank the authors for referencing the manuscript by Vesel et al, which provides an alternative perspective on a similar analysis. It would be helpful if the authors could comment on whether the R-M system unitigs that are associated with the differences in transformation (Table S6) correspond to the R-M systems that were identified and characterised in Vesel et al. It would be interesting to test whether DNA from different sources affected the efficiency of transformation of a bacterium harboring one of these R-M systems, but I sympathise with the view that such experiments are beyond the scope of this manuscript, which already contains extensive work. 

The authors state in their response that "As expected, both were significatively and negatively associated with transformation but phages in Ab and conjugative systems in Lp had a stronger inhibitory effect than restriction modification systems (Ab: Phages: effect=-0.97, p=4.3x10-3; RM: effect=-0.61, p=3.55x10-8 and Lp: Conjugative systems: effect=-0.76, p=1.4x10-4; RM: effect=-0.41, p=1.09x10-9)."

In both species, although the point estimate of the inhibitory effect of mobile elements is of a larger magnitude than that of R-M systems in these paired univariate analyses, the effect of R-M systems is more significant. Furthermore, no confidence intervals are provided for these estimates of the inhibitory effects, to determine whether they are significantly different from one another. Finally, there are biological reasons to expect that the distribution of mobile elements and R-M systems would not be independent of one another. Therefore it would be helpful to conduct a multivariate analysis of the effects of R-M systems and mobile elements, to quantify the relative contributions of the mobile element and R-M loci (including confidence intervals).

[2] The authors' analysis of plasmid diversity in their response is very interesting. However, such clarity is missing from the updated manuscript, which contains the confusing passage: "By blocking transformation, RocRp could also contribute to the preservation of chromosomal MGEs in positive epistatic interaction with the plasmid. In addition, plasmids could encode RocRp to avoid recombination with DNA entering by transformation from other partly homologous plasmids that could lead to deletion of non-homologous regions and/or create plasmid instability. Our results show that similar plasmids lacking RocRp do exist (Figure S11). In any case, these results strongly suggest that intragenomic conflicts between MGEs and natural transformation are not restricted to the effect of chromosome curing.".

I think it is worthwhile making it explicit that the rocRp gene is not widespread across multiple plasmids, but instead confined to a cluster of highly-related plasmids, given the RNA was described in Durieux et al, which had the title "Diverse conjugative elements silence natural transformation in Legionella species". A comparison of pLPL and pFFI102 suggests the RNA is within a cluster of genes that function to preserve the mobile element (e.g. an anti-restriction protein, a toxin-antitoxin system), and an integrase (plpl0043). As rocRp is shared with ICEs in other Legionella species (where chromosomal curing is a valid reason for carrying the RNA), it may be that this is a gene cluster for preserving conjugative elements that has been inserted once into the ancestor of the plasmids that carry rocRp through site-specific recombination. As it has only entered the L. pneumophila population once, it might be that rocRp actually has no direct role in benefitting the plasmid, but instead may be hitchiking on function of the other genes that do provide an advantage to the replicon (e.g. that encoding the anti-restriction protein). Furthermore, it may be that rocRp prevents homologous recombination between related plasmids that may delete the anti-restriction gene. However, given the absence of multiple parallel acquisitions of the rocRp gene by different plasmids, there is currently no direct evidence of rocRp itself directly benefitting the plamids, and therefore chromosomal curing may remain a valid explanation for its evolution. Nevertheless, it is certainly important to think about the possibilities for fresh ideas.

Minor comments :-

- The Discussion now includes the passage, "Transformation itself may be costly under certain circumstances and favour for a short term bacteria that lost it. But in the long term, the inability to take advantage of transformation may result in the purge of these lineages by natural selection." The authors themselves highlight this as potentially requiring editing for clarity, which I think would improve the work.

- Table S6a contains the phrase "conjugal protein", which should be "conjugative protein"

Reviewer #2:

I thank the authors for a much clearer second version of their manuscript, which addressed many points raised by 4 reviewers.

I concede to the authors that even though gubbins is not appropriate in the general case, it might be "less wrong" in the context of this work. The authors should perhaps mention this caveat (and perhaps a call to the community to implement an appropriate substitute method) in the Discussion and not just in the Methods.

While I appreciate the clarification regarding the horizontal lines in Figure 4A, I remain unconvinced that the authors are selecting an appropriate p-value threshold in their GWAS, even more so now that I can see a QQ-plot. For Acinetobacter in particular, the authors observe a flat line in the QQ-plot, and -log10 of the p-values that barely go over 6. This would lead me to expect no significant hits or just a handful of spurious ones; instead the authors report 1262 genes associated with increased or decreased transformability. That is ~35% of all genes, which is hard to believe and most certainly an artifact; unless the authors have a good explanation of why one third of the genome modulates this one phenotype. The authors should look again at their QQ plot and apply a more stringent threshold to come to a more realistic set of associated genes (if there are any). Applying a Bonferroni correction on the number of unique tested patterns (note that it is significantly less than the number of tested unitigs) is a recognized good practice which would provide a more sober result. I don't expect this to change much the results for Legionella, which has a few genes with very small p-values.

Reviewer #3:

Thanks to the authors for the thorough response to reviewers. Most comments from all four of us were adequately addressed. I would like to clarify two points I made that the authors still didn't quite address:

(1) Chicken-and-egg: In 3.6, the authors misunderstand my issue: Although MGE accumulation may slowly make strains less fit, this was not what I meant. Instead, the model itself presents two ways in which MGE-transformation can be in conflict that are difficult to resolve: (a) MGEs may mutate competence genes and make strains less transformable, in order to keep from being cured. This is the favored model of the authors, and certainly the odd comM "targeting" and RocRp RNAs are possible examples of this. However, significantly more broadly: (b) a simple point mutation in a competence gene that reduces transformability may lead to the accumulation of MGEs, irrespective of any direct effect of the MGEs themselves, since MGE "curing" role by transformation would be reduced. Indeed, there's a large body of experimental and theoretical work in higher eukaryotes that demonstrates how areas with reduced recombination (e.g. near centromeres, etc) accumulate MGEs, simply by virtue of having lower recombination rates. I think that the model the authors' propose is a bit wedded to the idea of certain MGEs directly impacting transformation, rather than the simply converse that they cannot be cured in non-transformable strains. Thus, many of the associations seen in their GWAS with MGEs (especially since these are abundant but weak) are merely associations and not causal. A role for MGE accumulation as a Muller's ratchet leading to extinction of low/non-transformable strains might indeed explain a lot about the distribution of competence within and among species. My point is that many of the MGE associations seen do not in any way suggest these are all affecting transformation rates.

(2) Luminescence correlations with transformation rates: The authors somewhat address me and Reviewer #4's concerns, but the dilution experiment only says that the technical method is reasonable and the use of known non-transformable isolates for baseline correction does not address the main issues we raised about natural genetic variation in competence induction, expression of the luc marker, or growth. This would require a handful of genetically distinct isolates with luminescence values across the range of measured values to actually have transformation experiments conducted. The authors did further elaborate why they think their values are correlated to transformation rates, but they should acknowledge that there's probably a fiar amount of "slop" in the values.

Reviewer #4:

All my comments have been addressed in a satisfactory manner.

---

## [Editor Report · Decision Letter 3]

27 Aug 2024

Dear Dr Mazzamurro,

Thank you for the submission of your revised Research Article "Intragenomic conflicts with plasmids and chromosomal mobile genetic elements drive the evolution of natural transformation within species" for publication in PLOS Biology. On behalf of my colleagues and the Academic Editor, Arjan de Visser, I'm pleased to say that we can in principle accept your manuscript for publication, provided you address any remaining formatting and reporting issues. These will be detailed in an email you should receive within 2-3 business days from our colleagues in the journal operations team; no action is required from you until then. Please note that we will not be able to formally accept your manuscript and schedule it for publication until you have completed any requested changes.

IMPORTANT: I've asked my colleagues to include the following requests among their own: 1. Many thanks for supplying the data underlying the Figs in the zipped supplementary folder. However, these files were in an odd format (HTML?) and would not open in Excel; Please could you submit them as separate .xslx files, renaming Tables S8, S9, S8_Data.xslx, S2_Dalta.xlsx, etc.? 2. Please change the corresponding call-outs in the main Fig legends from "Table S8," Table S9" to Data S8, Data S9, etc., and include analogous call-outs to supp data files in the supplementary Fig legends.

Sincerely,

Roli Roberts

Senior Editor

PLOS Biology

rroberts@plos.org